# Transcriptome-wide identification of transient RNA G-quadruplexes in human cells

Sunny Y. Yang[1], Pauline Lejault[2], Sandy Chevrier[3], Romain Boidot [3], A. Gordon Robertson[4], Judy M.Y. Wong [1] & David Monchaud [2]

Guanine-rich RNA sequences can fold into four-stranded structures, termed G-quadruplexes (G4-RNAs), whose biological roles are poorly understood, and in vivo existence is debated. To profile biologically relevant G4-RNA in the human transcriptome, we report here on G4RP-seq, which combines G4-RNA-specific precipitation (G4RP) with sequencing. This protocol comprises a chemical crosslinking step, followed by affinity capture with the G4-specific small-molecule ligand/probe BioTASQ, and target identification by sequencing, allowing for capturing global snapshots of transiently folded G4-RNAs. We detect wide-spread G4-RNA targets within the transcriptome, indicative of transient G4 formation in living human cells. Using G4RP-seq, we also demonstrate that G4-stabilizing ligands (BRACO-19 and RHPS4) can change the G4 transcriptomic landscape, most notably in long non-coding RNAs. G4RP-seq thus provides a method for studying the G4-RNA landscape, as well as ways of considering the mechanisms underlying G4-RNA formation, and the activity of G4-stabilizing ligands.

[1] Faculty of Pharmaceutical Sciences, University of British Columbia, Pharmaceutical Sciences Building, 2405 Wesbrook Mall, Vancouver, BC V6T 1Z3, Canada. [2] Institut de Chimie Moléculaire (ICMUB), UBFC Dijon, CNRS UMR6302, 9, Rue Alain Savary, 21078 Dijon, France. [3] Platform of Transfer in Cancer Biology, Centre Georges-François Leclerc, BP 77980, 1, Rue Professeur Marion, 21079 Dijon, France. [4] Genome Sciences Center, BC Cancer Agency, 570 W 7th Ave, Vancouver, BC V5Z 4S6, Canada. Correspondence and requests for materials should be addressed to J.M.Y.W. (email: judy.wong@ubc.ca) or to D.M. (email: david.monchaud@cnrs.fr)

The biological functions and cellular regulations of RNAs are dependent on their secondary and tertiary structures[1,2]. RNAs can adopt intricate bulged, stem–loop structures involving duplex-, triplex-, and quadruplex-RNA motifs[3,4]. G-quadruplexes (G4s) are structures formed by Hoogsteen bonding of four guanines to form planar guanine quartet (G-quartet) units, which π-stack on each other, to assemble into columnar four-stranded structures with the central cavity stabilized by monovalent cations (i.e., $K^+$, $Na^+$). G4 folding is spontaneous in vitro and results in a highly stable structure. While both single-stranded DNA and RNA can fold into G4s, the latter is less studied[5,6], even though G4 formation in RNA molecules is generally more stable, and RNA molecules can fold more readily due to their predominant single-stranded nature in vivo.

Formation of G4-RNA has recently been implicated in key RNA metabolism events, including the regulation of RNA processing and translation[7,8]. To better understand the roles of G4-RNA in cell biology, there has been a strong interest in mapping the distribution of G4-RNAs within the human transcriptome. However, to date, in vitro and in vivo evidence appear to be contradictory, limiting interpretation of the relevance of G4-RNAs[9–12]. In vitro transcription experiments suggest that opportunities exist for G4-RNA formation during co-transcriptional folding of nascent RNA[13]. Kwok et al.[9] used reverse transcription (RT)-stalling coupled with next-generation sequencing to map thousands of G4-RNA sites in vitro, showing widespread potential G4-forming sites within the human transcriptome. In contrast, Guo and Bartel[10] showed through in vivo DMS-mediated RNA modification with RT-stall-sequencing that G4s are nearly entirely in an unfolded state in vivo in mammalian cells. This observation is surprising since it has widely been assumed that G4-RNAs are formed in vivo, at least transiently. Conversely, evidence for in vivo G4 formation has been provided by cellular imaging studies using G4-specific antibodies and probes[11,12]. To reconcile these results, we hypothesized that G4-RNAs must be able to form, at least transiently, in live human cells, and that the identity of these G4-RNAs may provide valuable insights into their regulatory mechanisms and functions. However, because none of the above methods are suitable for capturing transient G4s, an alternative approach was needed.

Here, we report on a small-molecule-based approach to assessing the existence and the identity of transient G4-RNAs in the human transcriptome. We designed a biotinylated version of a previously characterized G4-specific ligand/probe, template-assembled synthetic G-quartet (TASQ), which self-assembles into a synthetic G-quartet upon association with a G4 target through end-quartet stacking. We developed a protocol, G4RP-seq (G4-RNA-specific precipitation and sequencing) using BioTASQ to capture G4-RNAs from human breast cancer cells in log-phase growth. Using this protocol to characterize in vivo transcriptomic landscapes, we showed that more G4s are present in gene transcripts that are GC-rich and have higher densities of predicted G4 motifs. We also evaluated G4 ligand-induced changes to the G4-RNA landscape following treatments with the G4 ligand, BRACO-19 or RHPS4, showing both similarities and differences in their respective induction profiles. Our data show that G4-RNAs can be ligand-induced in diverse RNA entities that include long non-coding RNAs; further, differential G4-RNAs induced by G4 ligands suggest that specific G4 structure–ligand interactions could be exploited.

## Results

**BioTASQ selectively captures G4 targets in vitro.** To support affinity purification and identification of functional transcriptomic G4-RNA targets, we added a biotin tag to the biomimetic quadruplex ligands known as TASQ (Fig. 1a)[14,15], known for their high G4-selectivity and, for some of them, their ability to track G4-RNAs in live cells (N-TASQ)[12,16]. Detailed synthesis and characterization of the biotinylated TASQ, or BioTASQ, can be found in Supplementary Figure 1 and Methods. We first evaluated the G4-interacting properties of BioTASQ via a fluorescence resonance energy transfer (FRET)-melting assay (Supplementary Figure 2)[17] and electrospray ionization mass spectrometry (ESI-MS) analyses (Supplementary Figure 3)[18]. FRET-melting experiments were performed with BioTASQ against a panel of dual-labeled nucleic acid sequences that included: (a) three G4-DNAs (F-Myc-T and F-kit-T, found in the promoter regions of *MYC* and *KIT* gene, respectively, and F21T, the human telomeric sequence);[19,20] (b) one G4-RNA (F-TERRA-T, the human telomeric transcript);[21,22] and (c) one duplex-DNA as a control (F-DS-T) (Supplementary Figure 2). Introduction of a biotin tag negatively impacted G4-affinity of BioTASQ (further confirmed by ESI-MS measurements) compared to the non-biotinylated parent [PNA]DOTASQ. However, it did not affect the ability of BioTASQ to discriminate G4s over duplexes[14,23].

Next, we assessed whether BioTASQ could efficiently capture G4s from solutions in vitro[24]. We used fluorescein (F)-labeled nucleic acids since fluorescence signal measurements allow for convenient and sensitive detection of the ligand/G4 assemblies. We performed these experiments with: (a) three G4-DNAs (F-Myc and F-SRC, two sequences found in the promoter regions of *MYC* and *SRC* gene, respectively[25,26], and F-22AG, the human telomeric sequence); (b) three G4-RNAs (F-TRF2 and F-NRAS, two sequences found in the mRNA of *TRF2* and *NRAS* gene, respectively;[27] and F-TERRA (the human telomeric transcripts); and (c) one duplex-DNA as control (F-Duplex). Labeled oligonucleotides (1 μM) were incubated with BioTASQ (20 μM) and streptavidin-coated magnetic beads (MagneSphere®, 25 μL). After overnight incubation at 25 °C, streptavidin beads were precipitated, the supernatant removed, and the beads resuspended in denaturing buffer before thermal denaturation (8 min at 90 °C). After separation from the beads (via centrifugation and magnet immobilization), fluorescent signals from the supernatant solutions were measured to quantify nucleic acid capture. Our results confirmed the efficiency of BioTASQ-mediated G4 capture (Fig. 1b) and showed that the level of G4s recovered was dependent on both the G4 nucleic acid type and topology. G4-DNA was enriched 4.1–20.7-fold, whereas G4-RNA was enriched between 10.9- and 23.8-fold when compared to controls, while duplex-DNA was not enriched. BioTASQ appeared to have stronger preferences for certain G4 topologies, as type I (or 'parallel')[28] G4s displayed better enrichment than type II (or "mixed-hybrid") G4s (with 16.6-, 20.7-, 20.6-, 10.9-, and 23.8-fold enhancement for F-Myc, F-SRC, F-TERRA, F-TRF2, and F-NRAS, respectively, versus 4.1-fold enhancement for F-22AG). The preference of BioTASQ for type I G4, which displays accessible external G-quartets, is expected, given that TASQs are sterically demanding ligands that require accessible, loopless G-quartets for binding G4 targets efficiently. This property represents a limitation to the use of BioTASQ for G4 detection, especially for G4-DNAs, which have higher conformational diversity than G4-RNAs. We have not yet tested the affinity of BioTASQ on the recently reported antiparallel G4-RNA[29]. While the cellular prevalence of G4-RNA with antiparallel topology remains to be established, they may provide key insights into the topological preference of TASQ ligands. We confirmed that the streptavidin bead/BioTASQ system did not extract duplex-DNA (0.7-fold). We further confirmed G4 selectivity of BioTASQ via competitive pull-down experiments, which we performed with

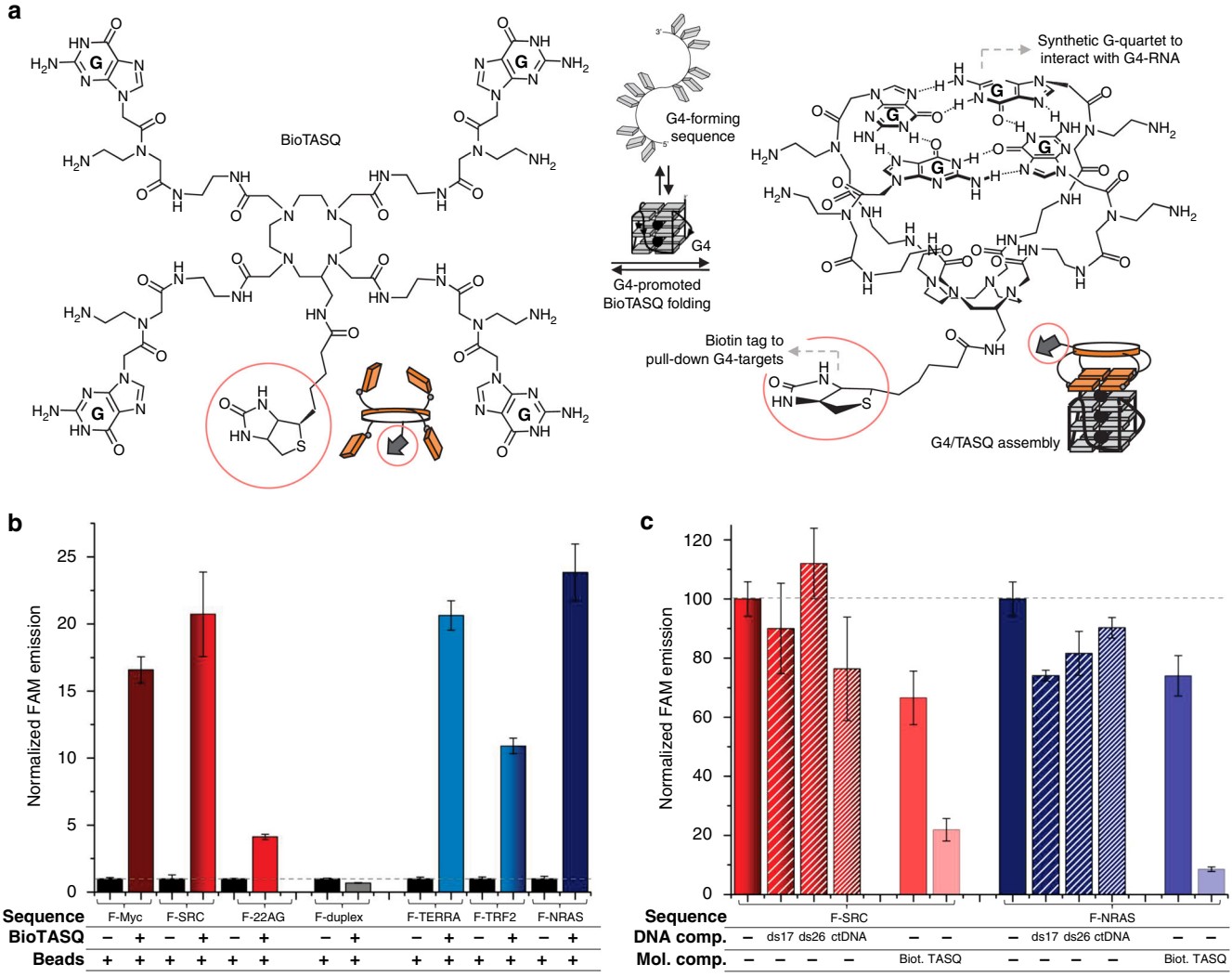

**Fig. 1** Characterization of G4-specific affinity of BioTASQ. **a** Structure of BioTASQ displaying a biotin affinity tag (red circles), and schematic representation of its open (left) and closed, quadruplex-associated conformation (right), in which the intramolecular G-quartet is formed. Schematic representation of a guanine-rich RNA sequence (guanines as gray squares) in its unfolded, random-coil and folded G4 structure. **b** Fluorescence analysis of pull-down experiments carried out with (1) FAM-labeled oligonucleotides (1 μM): either G4-DNA (F-MYC, F-SRC, and F-22AG), duplex-DNA (F-duplex) or G4-RNA (F-TERRA, F-TRF2, and F-RAS); (2) BioTASQ (20 μM); and (3) streptavidin-coated magnetic beads. **c** Competitive pull-down experiments performed with F-SRC and F-NRAS (1 μM), BioTASQ (20 μM) in the absence or presence of duplex-DNA competitors (ds17 or ds26, 20 μM) or DNA extracts (calf thymus DNA, ctDNA, 100 μM), or of molecular competitors (biotin (or Biot.), 80 μM or $^{PNA}$DOTASQ (or TASQ), 10 μM). All experiments were done in triplicates. Error bars represent SD

F-SRC and F-NRAS (1 μM) in the presence of an excess of duplex-DNA (either ds17 or ds26, 20 μM) or DNA extracts (calf thymus DNA, ctDNA, 100 μM, expressed in base pairs). The capture efficiencies of the fluorescently labeled G4-RNA were not significantly affected by an excess of synthetic duplexes (90–112% with F-SRC, 74-82% with F-NRAS) or with DNA extracts (76% and 90% with F-SRC and F-NRAS, respectively). We also tested BioTASQ/streptavidin association with an excess of either biotin (80 μM, to compete with streptavidin interaction) or $^{PNA}$DO-TASQ (10 μM, to compete with G4 interaction) (Fig. 1c). Together, these results show the strong ternary interaction between G4s, BioTASQ, and streptavidin (beads), which provided the basis for the development of our G4RP protocol (described below).

**G4RP isolates G4-RNA targets from human cell extracts**. After confirming that BioTASQ could interact with and capture G4s in vitro, we then assessed whether it could capture G4 targets

from human cell extracts. For this, we developed the G4-RNA-specific precipitation (G4RP) protocol, a modified version of the commonly used RNA-immunoprecipitation (RIP) protocol[30]. MCF7 cells were first crosslinked with formaldehyde to halt biological processes and stabilize transient structural interactions. Harvested cells were then sonicated briefly to release cellular content. Cell lysates were incubated with a high concentration of BioTASQ (100 μM) overnight (Supplementary Figure 3C) before affinity purification with magnetic streptavidin beads.

We first used RT quantitative PCR (RT-qPCR) with gene-specific primers to confirm the efficiency of the G4RP protocol[31]. G4RP-qPCR analysis of RNAs extracted from the BioTASQ-enriched fractions showed that non-specific binding was negligible (black bars) (Fig. 2a, b) while demonstrating the enrichment of two known G4-forming mRNAs, i.e., *VEGFA* and *NRAS*[27] (gray bars) (Fig. 2a, b).

To confirm bona fide G4 formation in target mRNA sequences, we collected MCF7 cell extracts following treatments with two well-established G4-stabilizing ligands, BRACO-19

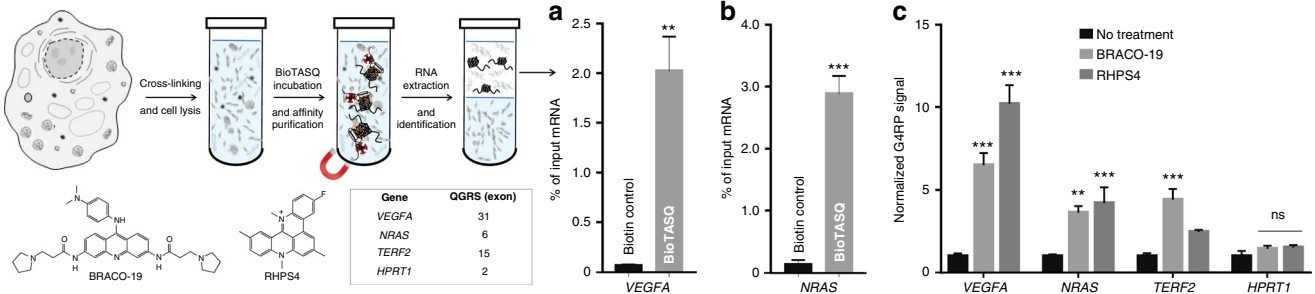

**Fig. 2** Isolation of G4 targets from human cell extracts using G4RP. (left) Schematic representation of G4RP protocol. G4RP signals of biotin control versus BioTASQ through RT-qPCR quantification of **a** *VEGFA* and **b** *NRAS* mRNA levels in untreated MCF7 cells. **c** Changes induced by BRACO-19 and RHPS4 measured by the fold change of G4RP signal in *VEGFA*, *NRAS*, *TERF2*, and *HPRT1* mRNA. Values are normalized to their individual untreated controls. Three biological replicates were used for the quantifications. Student's *t*-test and two-way ANOVA were performed. *p*-Values: *$p < 0.05$, ** $p < 0.01$, and ***$p < 0.001$. Error bars represent SEM

and RHPS4[32,33]. We chose the treatment ligand doses to be between the respective $IC_{15}$ and $IC_{25}$ (doses that are growth inhibitory in 15–25% of the cells; BRACO-19: 5 μg/mL and RHPS4: 1.5 μM), as determined by growth kinetics profiling of MCF7 in the presence of G4 ligands, compared with vehicle controls (Supplementary Figure 4). Treatment with G4 ligands significantly increased G4RP-qPCR signals by 6.5-fold, 3.6-fold, and 4.4-fold in BRACO-19-treated cells and by 10.2-fold, 4.2-fold, and 2.5-fold in RHPS4-treated samples in three selected G4-rich regions, *VEGFA*, *NRAS*, and *TERF2*, respectively, compared to the untreated control (Fig. 2c; Supplementary Figure 5), demonstrating that BioTASQ can specifically enrich for G4-containing RNA sequences. *HPRT1* was selected as an unstructured RNA control, as this housekeeping mRNA is expected to have low G4-forming potential[34]. Neither BRACO-19 nor RHPS4 treatment induced a significant change in BioTASQ-captured *HPRT1* signals, confirming that these G4 ligands were selective for G4-rich targets. Collectively, these results indicate that the G4RP protocol is suitable for the purification and identification of G4-containing RNAs from human cell extracts, as well as the quantification of the G4 ligand-induced changes.

We performed G4RP in samples that were crosslinked to preserve the transiently formed cellular G4s while preventing the induction of G4 formation in vitro, as crosslinked nucleic acids would be in an immobilized state. We used the reported formaldehyde concentration and crosslinking conditions where over 90% of nuclear DNA are immobilized[35] and anticipated that the crosslink efficiency for cellular RNA to be similar. As controls and to illustrate the importance of this crosslinking step, G4RP was performed in non-crosslinked samples. We selected three targets from the top- and bottom-ranked transcripts obtained with our G4RP-seq results (see below). BioTASQ enrichment of these targets was quantified using RT-qPCR and compared between the non-crosslinked and crosslinked samples (Supplementary Figure 6). We observed a loss of difference in BioTASQ enrichment in the non-crosslinked samples, despite the overall higher signals. These increased signals in non-crosslinked samples were likely G4s that were formed in vitro, due to the high concentration of BioTASQ, arguing that the crosslinking step is necessary to immobilize cellular RNA structures and to minimize the effects of in vitro G4 formation and destabilization through the biochemical evaluation steps. As naked, non-crosslinked RNA targets are susceptible to the in vitro G4 stabilization effects of high concentration of BioTASQ during the incubation steps, the true differences between the in vivo transient levels of the top and bottom ranking targets would be masked.

**G4RP-Seq identifies transcriptome-wide transient G4-RNAs.** To survey the baseline in vivo G4 transcriptomic landscape, we performed G4RP followed by sequencing (G4RP-seq) in human breast cancer cells harvested at log-phase growth. Due to the crosslinking step in the G4RP protocol, we expected to capture global levels of transient G4s in the transcriptome. Notably, as G4 ligand treatment resulted in gene expression changes in human cells[36], we needed to account for these input differences; therefore, an internal input control was included for each treatment condition. To enrich for diverse G4-forming RNAs, we elected to remove ribosomal RNA targets at the cDNA library preparation step, as rRNA constitute over 80% of total cellular RNAs. Ribo-depletion instead of poly-A selection was used, as we anticipated that non-poly-A RNAs, including many non-coding RNAs (ncRNAs), not only harbor but contribute to a substantial proportion of cellular G4s[37]. For the sequencing analysis, two comparisons were required: BioTASQ versus input (which gives normalized global levels of transient G4s for a specific transcript as Enrichment Score (ES)), and G4 ligand-treated versus untreated cells (given as Enrichment Score Change (ΔES)) (Supplementary Figure 7). After mapping reads to the hg19 reference human genome assembly with HISAT2[38], normalization and differential gene expression analyses were performed with HTSeq/DESeq2 pipeline[39,40], by comparing the BioTASQ-enriched samples with the corresponding inputs[41] (Supplementary Data 1). Of note, the relative enrichment levels by BioTASQ (given as ES) were not direct quantitative readouts of G4 formation but instead indicate the relative propensity of a specific transcript to fold into G4. ES values allow ranking of transcripts by relative G4-folding status under specific experimental conditions.

We observed BioTASQ enrichment of many gene transcripts, suggesting the existence of widespread G4-RNAs, at least transiently, in live human cells. This observation was expected since we were capturing a snapshot of the G4-RNA landscape in which some G4-rich sequences were in folded states, while others were in unfolded states. While the sequencing depth was not high enough to detect subtle changes in individual G4-forming sequences, we were able to confidently determine gene-level changes by focusing on a subset of highly abundant transcripts, filtered by high normalized mean read counts (as an estimate of relative expression level). We compared ES for each condition to generate initial lists of gene transcripts filtered by a minimal abundance threshold (>50 normalized mean read counts) (Supplementary Data 1). To ensure high-confidence hits with substantive changes, we further filtered the list and included only those with high transcript read abundance (≥500 mean read counts; approximately the top 5% of the list), which we used for

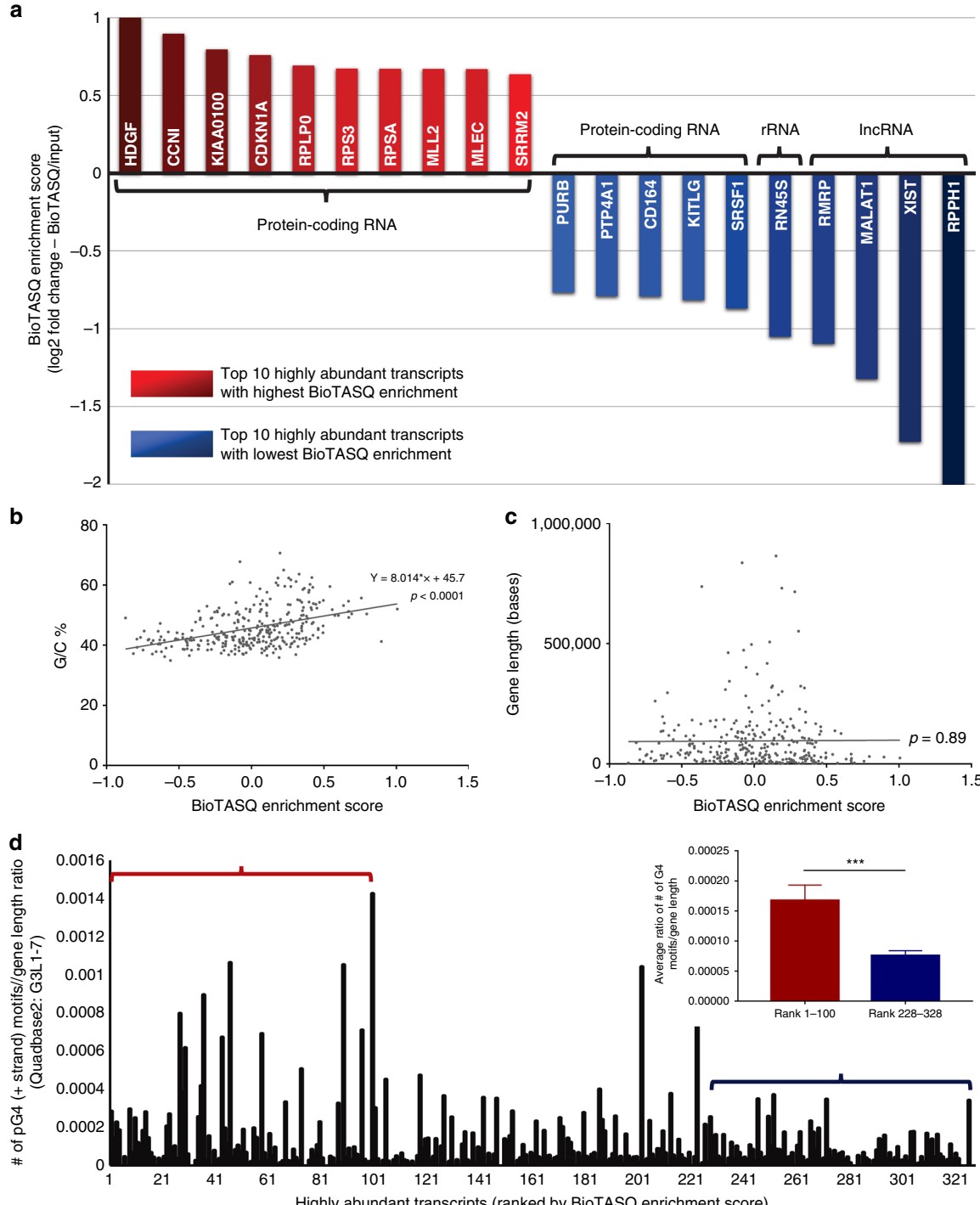

**Fig. 3** Characterization of the baseline level of G4-RNA landscape using G4RP-seq. **a** Top 10 highly abundant transcripts (filtered by at least 500 base-read counts) with the lowest BioTASQ enrichment (blue bars, normalized to the input in the untreated sample) or with the highest BioTASQ enrichment (red bars) ranked by Enrichment Scores. **b** Regression plot of BioTASQ-Enrichment Score for each transcript versus its corresponding G/C content ($R^2 = 0.187$, $p < 0.001$, significant non-zero relationship). **c** Regression plot of BioTASQ-Enrichment Score for each transcript versus its corresponding gene length ($R^2 = 0.00005$, $p = 0.89$, non-significant relationship). **d** Number of pG4 motifs (calculated by Quadbase2 using mid stringency G3L1-7) to gene length ratio plotted against the subset of highly abundant transcripts ranked by their BioTASQ enrichment. The bar graph (inset) shows the average pG4 motif/gene length ratio between the top 100 ranked transcripts versus the bottom 100 ranked transcripts ($p < 0.001$, significant difference, Student's $t$-test)

downstream analyses. Gene transcripts in this filtered list were then ES-ranked for downstream analyses (Supplementary Data 1D).

ES, a gene-specific ratio of the BioTASQ signal normalized to the corresponding input signal, moderately positively correlated with their respective G/C content (Pearson correlation = 0.43, $p < 0.0001$) (Fig. 3b). ES was uncorrelated to gene length (Pearson correlation = 0.02, $p = 0.89$) (Fig. 3c). To evaluate whether the ES is related to the density of potential G4 sequences, Quadbase2[42] was then used to assess the number of predicted G4 (pG4) motifs

(i.e. sequences with the canonical G3L1–7 G4 sequence) in each gene transcript. The ratios of the number of pG4s to gene length were calculated and then plotted against the transcripts ranked by their ES (Fig. 3d). Comparison of the ratios between the top and bottom 100 ES-ranked transcripts showed significantly higher values (2.2-fold difference) for top-ranked transcripts. Results from gene ontology analysis of the top and bottom 100 ES-ranked transcripts are summarized in Supplementary Table 1[43]. Together, our bioinformatics analyses confirmed that transcripts with higher ES tend to have higher G/C content and higher pG4 density.

We also found that highly expressed lncRNAs had some of the lowest relative G4 levels, as shown by their low ES (Fig. 3a). These bottom-ranked transcripts included well-known lncRNAs such as *MALAT1*, *RPPH1*, *RMRP*, and *XIST*. The presence of *MALAT1* on this list was unexpected, as it has been previously reported as a G4-forming lncRNA in vitro[9,10]. Interestingly, residual rRNAs (that escaped from the ribodepletion step during library preparation) were also among the most depleted within this list.

**G4RP-seq identifies ligand-induced changes in G4-RNA profiles.** We evaluated ligand-induced changes in the G4-RNA landscapes by applying G4RP-seq to samples treated with G4 ligands and calculating an Enrichment Score Change (ΔES) from the ratio of treated versus non-treated samples. By filtering the initial list ΔES > 1.75, we found BioTASQ enrichment in 251 and 463 gene transcripts to be highly induced by BRACO-19 and RHPS4, respectively (Supplementary Data 1B–C). Results from gene ontology analysis performed on the lists of top genes are summarized in Supplementary Table 2[43].

Among the list of gene transcripts with a high ligand-induced increase (ΔES > 1.75) and at least 500 mean read counts, the lncRNAs *MALAT1*, *RPPH1*, and *XIST* were highly ranked in both BRACO-19- and RHPS4-induced gene lists (Fig. 4a). Some RNAs previously reported to harbor G4s in vitro were also identified, including *NEAT1* and *NRAS*[9,27]. We found no correlation between the ligand-induced ΔES and the read counts of the transcripts (Pearson correlation = 0.02, p = 0.69) (Supplementary Figure 8). The ligand-induced ΔES of each transcript was compared to the corresponding G/C content of the transcripts, which interestingly showed a negative correlation (Fig. 4b). When the transcripts were ordered by ΔES, the pG4 density appeared to be distributed toward lower scores (1.7-fold and 2.7-fold difference between the average ratio of top and bottom 100 ranked transcripts for BRACO-19 and RHPS4, respectively) (bottom panel) (Fig. 4b). Overall, our observations suggest that transcripts with higher pG4 density were more likely to be captured in a folded state in the absence of ligands, resulting in ΔES being lower due to the higher baseline level of G4 formation. In contrast, transcripts with lower pG4 density were more likely to be unfolded in the absence of ligands and to have their G4 structures stabilized in the presence of ligands, leading to a higher ΔES.

When we compared the absolute number of pG4 motifs (i.e. without normalization to gene length) between the three treatment conditions ranked by their respective ES, we observed differential changes in pG4 profiles between the two G4 ligand treatments (Fig. 4c). pG4 scores generated using different stringency of searches (G2L1–10, G3L1–5, and G3L1–7) showed similar trends (Supplementary Figure 9). BioTASQ-captured targets generated from BRACO-19-treated samples exhibited higher levels of pG4-dependent enrichment regardless of the search stringency, conceivably due to the broader range of intramolecular G4s (longer loops, 2-quartet G4s, etc.) stabilized by this ligand. On the other hand, targets generated from RHPS4-

treated samples showed lower levels of pG4-dependent enrichment and a pG4-dependency could only be observed when the plots were obtained using G4 motif searches with the lower stringency. We reasoned that RHPS4-binding preference could be selective towards sequences with lower numbers of G4 motifs or highly specific G4 sequences that are less prevalent within the transcriptome. However, we cannot rule out the possibility of intermolecular G4s, as computational algorithms are currently unable to predict these structures. Given this caveat, the lack of pG4-dependent enrichment in samples treated with RHPS4 could be partially explained by a preferential ligand-induced stabilization of intermolecular G4s. The overlap between the gene lists for the two G4 ligand treatments demonstrated that they have differential G4-induction profiles, in agreement with their differential in vitro G4-structure-specific binding profiles (Fig. 4d)[44].

We further validated our findings from G4RP-seq by performing qPCR on separate G4RP samples obtained with biological repeat experiments, using primers specific for the top three common lncRNA targets: *MALAT1*[45], *XIST*[46], and *RPPH1*[47]. Fold change differences between G4RP-qPCR and G4RP-seq were seen and expected due to the small qPCR region amplified. Nevertheless, the qPCR results were consistent with those obtained from sequencing, confirming that these lncRNAs were targets of G4 ligands (Fig. 4e; Supplementary Figure 10). Circular dichroism (CD) and thermal differential spectra (TDS) analyses[48] of the three selected pG4 regions of *MALAT1*, *XIST*, and *RPPH1* were consistent with the formation of parallel-type G4 structures in vitro in the selected pG4 motif sequences extracted from these genes (Supplementary Figure 11). In summary, we observed that the BRACO-19 and RHPS4 treatments in MCF7 cells similarly induced G4 stabilization in several highly expressed lncRNA targets, but the treatments also displayed distinct ligand specificity towards other RNA targets.

## Discussion

It has been widely assumed that G4s must be formed in vivo, at least transiently, due to the high structural stability of G4 nucleic acids and the favorable intracellular potassium concentration. The extent to which G4-RNA formation occurs in vivo in human cells is debated[49], since it has been recently reported that G4s are globally unfolded in the mammalian transcriptome[10]. To address this conundrum, we developed and reported here on G4RP-seq, a protocol that provides evidence supporting the existence of transient G4-RNAs in the in vivo human transcriptome. We also used G4RP-seq to generate readouts of the changes in transcriptome-wide G4-RNA landscape upon treatments with G4 ligands.

Our first key finding of global transient G4-RNA formation suggests an alternative perspective on Guo and Bartel's[10] lack of detected G4-RNAs by DMS/RT-profiling, an observation that can be interpreted in two ways: either G4s are seldomly formed in vivo, or G4s are formed but are unfolded quickly by destabilizing mechanisms such as through the actions of helicases. DMS/RT profiling may be reporting on an unfolded G4 landscape since it measures the biological endpoint in which all G4s are eventually unfolded. A recent study corroborated this by demonstrating that dynamic folding and unfolding of some G4-RNAs in live cells could occur within seconds[50]. We postulated that even if the G4 equilibrium heavily favors an unfolded G4 state, global snapshots of transiently folded G4s could still be captured by a chemical crosslinking step. Our observation of widespread G4-RNAs using a snapshot approach, combined with Bartel's observation of endpoint globally unfolded G4-RNAs, suggest that in vivo G4-RNAs can form continuously and are rapidly resolved

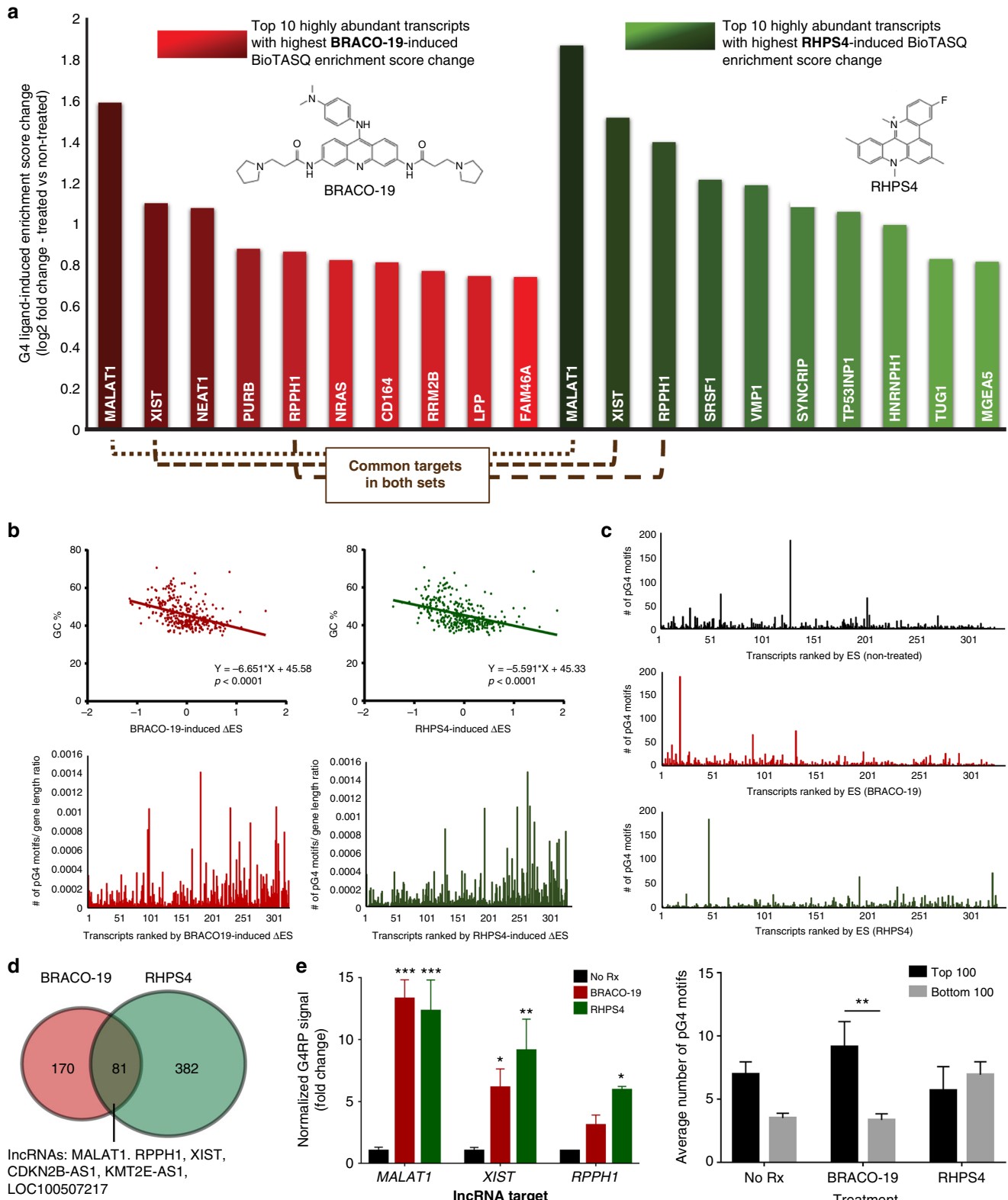

to an unfolded state. Further, we anticipate that the equilibrium between an RNA's transient folded and unfolded states can be influenced by competing factors such as the transcription rate of the RNA, the availability of helicases[51], the chaperone activities of RNA-binding proteins[52], and the structural stabilization by G4 ligands[53] (Fig. 5). Importantly, the effectiveness of capturing these transient G4-RNAs within transcripts, as measured by BioTASQ

enrichment, is correlated with their G/C content and pG4 density. This observation suggests that gene transcripts with a higher density of pG4 motifs would have a higher probability of having at least one of these motifs captured in a folded state with our crosslinking snapshot approach. It is currently undetermined whether such transient G4s have any biological roles or are simply stochastic passenger events that arise from the chemical

**Fig. 4** Characterization of the ligand-induced changes in the G4-RNA landscape. **a** Top 10 highly abundant transcripts with highest fold increase in BioTASQ enrichment (ranked by Enrichment Score Change) for BRACO-19 (red) and RHPS4 (green). Common targets that were ranked highly for both ligands are indicated in brown dashed lines. **b** (Top) Regression plot of BioTASQ-Enrichment Score Change for each transcript versus its corresponding G/ C content (BRACO-19: $R^2 = 0.16$, $p < 0.0001$; RHPS4: $R^2 = 0.16$, $p < 0.0001$, significant non-zero relationship). (Bottom) Number of G4 motifs (calculated by Quadbase2 using mid stringency G3L1–7) to gene length ratio plotted against the subset of highly abundant transcripts ranked by their BioTASQ-Enrichment Score Change. Left panel is BRACO-19-induced changes and right panel is RHPS4-induced changes. **c** Absolute number of pG4 motifs of highly abundant gene transcripts ranked by BioTASQ-Enrichment Score for the three conditions: non-treated (black), BRACO-19 (red), and RHPS4 (green). (Bottom) Quantification of average number of pG4 motifs for top 100 and bottom 100 ranked genes for the three sets of conditions. **d** Venn diagram comparing top filtered BioTASQ-enriched gene lists for BRACO-19 and RHPS4. **e** G4RP-qPCR controls of the top lncRNA hits MALAT1, RPPH1, and XIST, in G4 ligand-treated (BRACO-19 or RHPS5) samples normalized to corresponding untreated controls. Three biological replicates were used for quantification. Two-way ANOVA was performed. $p$-Values: *$p < 0.05$, **$p < 0.01$, and ***$p < 0.001$. Error bars represent SEM

properties of guanines. Strikingly, lncRNAs appeared to avoid G4-formation and were found mainly in an unfolded state, in the absence of G4 ligands. The functions and interactions of lncRNAs with other molecules (DNA, RNA, and protein) are dependent on their folding into higher-order structures[54], and the formation of G4 may interfere with such folding. Consistent with this, a study on hnRNP F binding and G4 formation showed that hnRNP F ribonucleoprotein formation prevents G4 formation/reformation by sequestering the G-tracts in a single-stranded RNA state[55]. As the rate of protein-RNA complex formation is faster than that for G4 formation[56], the shift of cellular G4-RNAs toward an unfolded state could be partially due to interactions with RNA-binding proteins.

The second key finding from our report is the characterization of G4 ligand treatment effects on the G4-RNA folding state within the in vivo human transcriptome. While previous studies have characterized G4 ligand-induced changes for human genome-wide G4-DNAs and transcriptome-wide G4-RNAs in vitro[9,57], the effects of G4 ligands on transcriptome-wide G4-RNAs in vivo have not been reported. Here, we showed by the comparison of BioTASQ-enrichment profiles of BRACO-19 and RHPS4 that these two G4 ligands have both shared and distinct targets. Following ligand treatments, we found, paradoxically, that transcripts with lower pG4 density were relatively more enriched than those with a higher number of G4 motifs. There are two possible explanations for this. First, the relative increase in Bio-TASQ enrichment is less significant in high-pG4 transcripts than that found in low-pG4 transcripts, which have, by definition, a lower probability of being crosslinked in a folded state in the absence of ligands. Upon ligand treatment, stabilization of these previously unfolded pG4s within the low-pG4 transcripts significantly increased their BioTASQ enrichment. A second possibility is that rather than the density of G4 motifs, ligand-induced G4 stabilization may be more dependent on the ligand-binding affinity of individual G4 sequences, taking into account other confounding factors such as topology, flanking or loop sequences, and the presence of modified nucleosides. This second possibility is supported by our observation that the absolute number of pG4s within a transcript seemed to be more predictive of its ligand-induced G4-structure formation for BRACO-19, a broad spectrum, pan-G4-specific ligand, but not for RHPS4, a ligand with higher structural and sequence specificity. Given that the G4-binding modes of BRACO-19 and RHPS4 are different, their pG4 profiles are expected to be different.

While BRACO-19 and RHPS4 G4-profiles are distinct in several aspects, they also share common targets within the in vivo MCF7 transcriptome, most notably in the abundant lncRNAs. We anticipate that these targets are of interest, as they contain G4 targets that can be accessed and stabilized by both tested G4 ligands, suggesting that these G4s may be druggable. We propose that lncRNAs can spontaneously form G4s (e.g. *MALAT1*)[9,10], but that G4-formation is actively counteracted by the actions of

helicases and/or other RNA-binding proteins[51,52]. We anticipate that there may be a window of opportunity for ligand-induced G4-formation/stabilization between the initial creation of nascent RNA and the RNA–RNA/RNA–protein interactions that are part of the co-transcriptional maturation stage of the ribonucleoprotein complex[13] (Fig. 5). It is important to consider the folding/unfolding kinetics, since the balance may be shifted as ligand-mediated G4 stabilization outcompetes the G4-destabilizing factors (i.e. duplex structures[58], helicases[51], and RNA-binding proteins[52]). As dysregulation of lncRNAs has been implicated in various human diseases, including cancers, cardiovascular, and neurodegenerative diseases, we anticipate that targeting G4s within lncRNAs may present a valuable therapeutic strategy to alter the functions of these RNA entities[59].

The G4RP-seq protocol has certain limitations and, we anticipate, can be further improved. One concern is that Bio-TASQ is itself a G4 ligand and thus may alter the G4 landscape. However, as demonstrated, the chemical crosslinking step before BioTASQ binding serves to minimize the effects of BioTASQ-induced stabilization (Supplementary Figure 6), we contend that the G4RP protocol should provide a relatively unbiased readout of cellular G4-RNA. On the other hand, as we have stated earlier, BioTASQ preference toward parallel G4s could limit its capability in capturing rarer forms of G4-RNAs (i.e. antiparallel G4s). Additionally, the G4RP protocol uses formaldehyde as a cross-linking agent, and this is known to capture both direct and indirect RNA–RNA interactions;[60] it will be informative in future work to also include the use of other crosslinking agents to better characterize the G4-RNA interactome. Additionally, the sequencing depth in our work was insufficient to differentiate small changes in BioTASQ enrichment in low abundance transcripts and individual G4-forming sequences, and we anticipate that high-resolution sequencing in future studies will be necessary for a more complete mapping of potential transcriptomic G4 sites. Despite these limitations, we have shown here that G4RP-seq is useful for the broad identification of transient G4 structures and offers a snapshot view of the G4 landscape in live human cells. While using G4RP-seq alone cannot distinguish which competing factors (i.e. RNA-binding proteins or competing secondary structures) played more important roles at specific sites, data from G4RP-seq could be studied in combination with other functional genomic strategies (i.e. G4-ChIP-seq[61], RIP-seq[30], rG4-seq[9], DMS-seq[10], LIGR-seq[62]) to better characterize the interactions between G4 genome and transcriptome (collectively the G4ome).

In summary, our work provides a proof-of-principle for studying the mammalian G4-RNA landscape, and a method for studying the dynamics of in vivo transient G4-RNA under various biological conditions. Importantly, through G4RP-seq, we have also evaluated the mechanisms underlying the biological activity of G4 ligands. This opens exciting opportunities in which G4RP-seq, by providing transcriptome-wide views of G4 level

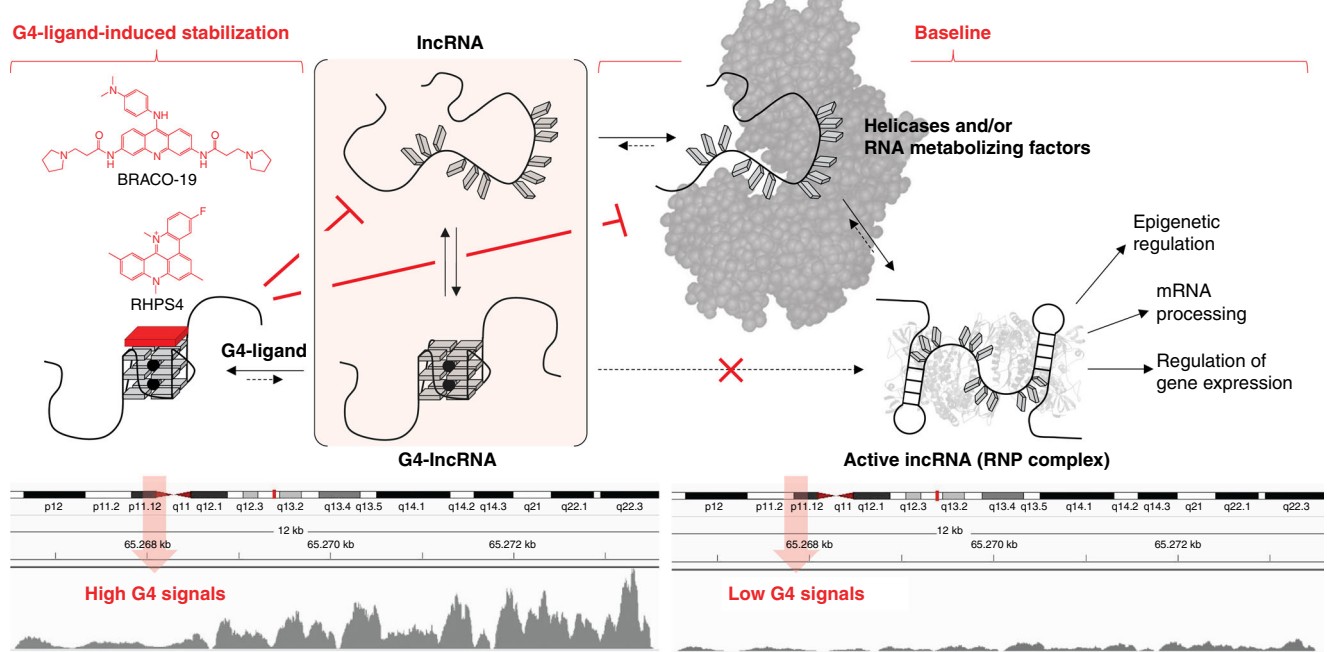

**Fig. 5** Proposed model of G4 structural equilibrium of lncRNAs in the presence of G4-stabilizing and G4-destabilizing factors. Guanine-rich RNAs in lncRNAs exist in equilibrium between single-strand and folded G4 state (center panel). At steady state (normal cell biology or untreated state), the nascent lncRNAs avoid G4-formation by the actions of G4-destabilizing factors such as helicases and RNA metabolizing factors (i.e., RNA-binding proteins) (right panel). G4-destabilizing factors may contribute to proper folding (including duplex structures) into the active form of lncRNA to mediate biological functions such as epigenetic regulation, mRNA procession, and gene expression regulation. The lack of G4 formation at steady state is shown by a lower relative level of G4RP-seq signal. However, in the presence of G4-stabilizing ligand (such as BRACO-19 and RHPS4), the nascent G-rich lncRNAs can be trapped in a folded G4 state due to G4-stabilizing ligands outcompetes the G4-destabilizing factors (left panel). This increase in folded G4 state is shown by an increase in the relative level of G4RP-seq signal

changes, could be used as a quantitative method in the iterative chemical design of new G4 ligands.

## Methods

**FRET-melting experiments**. The sequences of the oligonucleotides used herein are listed in Supplementary Table 3. The preparation of these sequences is described in the Supplementary Methods. Experiments were performed in a 96-well format using a Mx3005P qPCR machine (Agilent) equipped with FAM filters ($\lambda_{ex} = 492$ nm; $\lambda_{em} = 516$ nm) in 100 μL (final volume) of 10 mM lithium cacodylate buffer (pH 7.2) plus 10 mM KCl/90 mM LiCl (F21T, F-DS-T) or plus 1 mM KCl/99 mM LiCl (F-Myc-T, F-Kit-T, F-TERRA-T) with 0.2 μM of labeled oligonucleotide and 0, 1, 5, or 10 μM of BioTASQ. Competitive experiments were carried out with labeled oligonucleotide (0.2 μM), 10 μM BioTASQ, and increasing amounts (0, 15 and 50 equiv.) of the unlabeled competitor ds26. After a first equilibration step (25 °C, 30 s), a stepwise increase of 1 °C every 30 s for 65 cycles to reach 90 °C was performed, and measurements were made after each cycle. Final data were analyzed with Excel (Microsoft Corp.) and OriginPro®9.1 (OriginLab Corp.). The emission of FAM was normalized (0 to 1), and $T_{1/2}$ was defined as the temperature for which the normalized emission is 0.5; $\Delta T_{1/2}$ values are means of triplicates.

**In vitro pull-down assay**. The sequences of the oligonucleotides used herein are listed in Supplementary Table 3. The preparation of these sequences is described in the Supplementary Methods. The in vitro quadruplex capture experiments were performed in 250 μL final volume as follows: first, the streptavidin MagneSphere® beads (Promega) were washed three times with 10 mM lithium cacodylate buffer (pH 7.2) plus 10 mM KCl/90 mM LiCl (Caco.K) buffer. To this end, 200 μL of the commercial solution of beads were centrifuged, taken up in 200 μL of Caco.K, and this washing step is repeated three times. After the original solutions being reconstituted (as 250-μL mixtures in Caco.K), 25 μL of MagneSphere® beads were added to solutions comprising BioTASQ (20 μM) and FAM-labeled oligonucleotides (1 μM). The mixtures were stirred overnight at 25 °C; then, the mixtures were subsequently centrifuged (60 s at 8900 r.p.m.), the beads are immobilized (attracted by a magnet) and the supernatant was removed. The solid residue was resuspended in 240 μL of TBS 1× buffer, heated for 8 min at 90 °C (under gentle stirring 800 r.p. m.), and then centrifuged for 2 min (8900 r.p.m.). The supernatant was taken up for analysis (the beads being immobilized by a magnet), after being splitted in three wells (80 μL each) of a 96-well plate using a ClarioStar machine (BMG Labtech)

equipped with FAM filters ($\lambda_{ex} = 492$ nm; $\lambda_{em} = 516$ nm). Data were analyzed with Excel and OriginPro®9.1. FAM emission was normalized as follows: the FAM emission of the three control wells (without BioTASQ) was collected and normalized to 1; then, the FAM emission of the three wells comprising labeled oligonucleotides, BioTASQ, and beads were collected and compared to the control experiments. This allowed for a direct quantification of the BioTASQ capture efficiency. Competitive experiments were performed with BioTASQ (20 μM), labeled oligonucleotides (1 μM), and MagneSphere® beads (25 μL) along with the either nucleic acid competitors, ds17 (20 μM), ds26 (20 μM) or calf thymus DNA (CT-DNA, 100 μM in base pairs), or small-molecule competitors, biotin (80 μM) or $^{PNA}$DOTASQ (10 μM). All experiments were performed in triplicates.

**Cell line and culture**. Human breast cancer cells MCF7 was obtained from American Type Culture Collection (ATCC). The cells were culture in Dulbecco's modified Eagle's medium (Gibco) supplemented with 5% synthetic fetal bovine serum (FetalClone III; GE Life Sciences) and 100 U penicillin–streptomycin mixture (Gibco). Cells were incubated at 37 °C in a humidified, 5% $CO_2$ atmosphere-controlled incubator (HERAcell). Standard cell-culturing procedures were employed including aspiration and washing with phosphate-buffered saline (PBS, Gibco). Cells were trypsinized using Trypsin-EDTA (0.25%) (Gibco). Cell counting was performed using a Coulter Counter (Beckman Coulter).

**Dose–response profiling of G4 ligands**. MCF7 cells were seeded at 3000/well in a 96-well flat bottom plate. The cells were treated with a series of BRACO-19 or RHPS4 concentrations, made from serial dilutions. The cells were then monitored in the Essen Bioscience IncuCyte ZOOM live-cell monitoring system[63]. Phase confluency was used to measure cellular proliferation under a range of doses. The maximum and minimum achievable confluency values under these conditions were used for normalization. Three biological replicates were used to produce the dose–response curve. Doses between $LD_{15}$ to $LD_{25}$ were calculated from dose–response curve profiles (Supplementary Figure 4) and used for subsequent experiments.

**G4 RNA-specific precipitation (G4RP)**. MCF7 cells were seeded at $3.5 \times 10[5]$ cells per 10-cm dish before treatment with either vehicle (PBS), BRACO 5 μg/mL ($LD_{15}$), or RHPS4 1.5 μM ($LD_{25}$) for 72h. Cells were then crosslinked using 1% formaldehyde/PBS for 5 min at 25 °C and the crosslinking was then quenched with

0.125 M glycine for 5 min. Cells were scraped and resuspended in G4RP buffer (150 mM KCl, 25 mM Tris pH 7.4, 5 mM EDTA, 0.5 mM DTT, 0.5% NP40, RNase inhibitor (Roche), homebrew protease inhibitor cocktail). Cells were then sonicated using a Covaris m220 Ultrasonicator using default settings at 10% duty for 2 min. The sonicated fractions were then incubated with 100 μM BioTASQ (or 100 μM biotin for negative controls) overnight at 4 °C. Five percent of the sonicate was collected as input control. Ten micrograms of streptavidin-magnetic beads (Promega) was added and the extract was incubated for 2 h at 4 °C. Magnetic beads were then washed four times in G4RP buffer for 5 min. The beads were then incubated at 70 °C for 1 h to reverse crosslink. TRIZOL was then used to extract the RNA from the beads using the manufacturer's instructions.

**RT-qPCR.** The primer sets used for RT-qPCR are listed in Supplementary Table 4. Extracted RNA was reverse transcribed with Superscript III (Thermos) and random hexamer primers using the manufacturer's standard protocol to generate cDNA. cDNAs were quantified using 2× SYBR green mix (Bimake) with three technical replicates. $C(t)$ values of pull-down samples were normalized to the input control. Three biological replicates were used for all qPCR-based quantifications. Exon-spanning primers for quantifying mRNA levels were derived from Primerbank[31].

**G4RP sequencing (G4RP-seq) and analysis.** G4RP samples were DNAase-treated, briefly thermally fragmented and ribo-depleted using the Illumina Ribo-Zero rRNA removal kit. RNA library preparation was performed using the Illumina TruSeq RNA Library Prep kit by following the manufacturer's instructions. Two replicates (non-treated, BRACO-19 treated and RHPS4) along with an input for each condition were paired end sequenced at 2 × 75 bp using NextSeq 500. The sequenced reads were mapped to the human reference genome hg19 assembly using HISAT2[37]. Exon feature count and annotation to genes were performed using HTSeq[38]. DESeq2[39] was used for normalization and differential gene expression analyses. Further filtering and analyses were performed on Excel. The analysis workflow is outlined in Supplementary Figure 7. Initial gene lists were filtered by mean read count of >50 and fold change of >1.75 (log value of 0.8). The list of abundantly expressed genes were filtered by a mean read count of >500. GO analyses were performed using Enrichr[42]. Detailed list of gene transcripts can be found in Supplementary Data 1.

**Statistics.** Graphs were produced by Microsoft Excel or GraphPad Prism. Statistical testing for multiple groups of dataset were performed using one-way or two-way ANOVA, and multiple comparisons corrected by the Bonferroni's method. Statistical comparisons of the average of two groups were performed using two-tailed Student's t-test. Linear regression goodness of fit was determined from coefficient of correlation ($R^2$) or Pearson correlation ($R$), and non-zero slope significance was given as p-value <0.05. All p-values <0.05 were considered significant unless specified otherwise.

**Reporting Summary.** Further information on research design is available in the Nature Research Reporting Summary linked to this article.

## Data availability

All relevant data are available on request from the authors upon reasonable request. The sequencing data have been deposited into NCBI's Gene Expression Omnibus and are accessible at GSE112898. A reporting summary for this Article is available as a Supplementary Information File.

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

## Acknowledgements

This work, supported by the Agence Nationale de la Recherche (ANR-17-CE17-0010-01), the European Research Council (H2020-MSCA-IF-2016-750368), is part of the project Pharmacoimagerie & agents théranostiques supported by the Université de Bourgogne Franche-Comté and Conseil Régional de Bourgogne (PARI) and the European Union (PO FEDER-FSE Bourgogne 2014/2020 programs) and the Research Reinvestment Funds from the University of British Columbia. The authors are grateful to Isaline Renard and Claire Bernhard for the BioTASQ synthesis, Apolline Roux and Marc Pirrotta for the in vitro pull-down assays.

## Author contributions

S.Y.Y., J.M.Y.W., and D.M. designed the experiments. P.L. and D.M. performed in vitro experiments; S.Y.Y. and J.M.Y.W. performed cell-based investigations; S.C. and R.B. performed NGS experiments; S.Y.Y., J.M.Y.W., and G.R. analyzed the NGS data; S.Y.Y., G.R., J.M.Y.W., and D.M. interpreted the results and co-wrote the manuscript.

## Additional information

**Competing interests:** The authors declare no competing interests.

