## [Peer Review File · Nature Communications]

Reviewers' Comments:

Reviewer #1:

Remarks to the Author:

This manuscript from Yang et al describes the detection of RNA G Quadruplexes in fixed cells. The authors employ G4RP-Seq - pulldown of G Quadruplexes followed by deep sequencing - to do so. This work is a timely contribution to a unsolved problem of key importance to RNA biology - do G Quadruplexes form in live eukaryotic cells?

This work fits nicely in a niche between live-cell G4-Seq (recent work from the Bartel lab and others), in which they observed these structures were unfolded in eukaryotes but folded in prokaryotes, and fixed-cell work with antibody staining from the Balasubramanian lab and others in which they observe G Quadruplexes in eukaryotic cells. Here, the Wong and Monchaud labs propose G4RP-Seq can be used as a snapshot-type method for capturing transient G Quadruplexes in eukaryotes. The field is currently in flux right now and this certainly won't be the final word on G4 RNA folding in eukaryotes, but that's precisely why this is an important contribution that will influence thinking in the field. It is suitable for the broad audience of Nature Communications with only minor revisions and one or two additional in vitro experiments.

My suggestions are below:

Major:

It is claimed that the pulldown complex is ternary (avidin/BioTASQ/G4 RNA). However, it's not clear to me that this is the case. Given the authors' previous observation in their report of N-TASQ that the 1:1 N-TASQ/TERRA complex can interact further with another equivalent of N-TASQ (e.g., Figure 2B of their 2015 JACS article) one would expect that at least a quaternary complex (1 avidin:2 BioTASQ:1 G4 RNA) is possible. Furthermore, this likely would only be possible in some cases, providing another source of bias in the experiments.

Given the concentrations employed, it appears likely that the avidin molecules on the beads are loaded with multiple equivalents of BioTASQ, and quadruplexes that can interact with more than one BioTASQ at once will have longer bound lifetimes. I'm not insistent this be fully sorted out for a first report, but the authors seem to be most of the way there on an experiment that would be very helpful to readers. Doing some Job plots with soluble avidin-BioTASQ (or at least BioTASQ on its own) and a few G Quadruplexes of differing pulldown efficiency, and then correlating this to bound lifetime by binding fluorescently labeled quadruplexes to avidin beads with bound BioTASQ, washing, and monitoring the fluorescence of the supernatant over 2-3x the experimental timescale could provide some key information of value to readers in interpreting the data from this new technique.

Given the bias of the ligand employed for parallel G quadruplexes, some discussion (at least a more explicit acknowledgement to the reader) that this method may be missing some of the (rarer, but now known) antiparallel RNA G Quadruplexes would be helpful to those unfamiliar with the field.

Similarly, a discussion of their experience with TASQ type ligands exhibiting tighter binding to G quadruplexes with exposed quartets that lack occluding loops would be helpful in allowing the reader to manage expectations regarding the limitations of this technique.

Minor:

Concentrations are not clear. Oligonucleotides are reported as being present at 1e-6 M, but ctDNA is reported as being present at 100e-6M. Is this phosphate? Base pair?

Page 2, line 23 - this seems to conflate two phenomena. While many RNA G Quadruplexes are more stable than their DNA equivalents, this is a separate phenomenon from the fact that, in a cell, DNA is generally present paired to its complement, while RNA is single-stranded (or folded). Language should be adjusted.

Reviewer #2:

Remarks to the Author:

In the manuscript „Transcriptome-wide identification of transient RNA G-quadruplex in human cell“ by Yang et al. the authors claim the discovery of a novel technique to identify RNA-G-quadruplexes in vivo.

RNA G-quadruplexes have attracted a lot of attention in the last years. Recently, a publication of the Bartel group in Science revealed that under their experimental conditions RNA-G4s are mainly unfolded. To address the controversial discussion, if G4 structures do not exist at all or if they are just rapidly unfolded by protein in the cell the authors attempted to identify RNA-G4s in vivo.

Overall, the manuscript is well written, but the outlined experiments and results are not well presented and lack biological or medical relevance.

Major points

1. The experimental approach is well designed and interesting. However, it is not clear how an artificial ligand that is stabilizing RNA-G4 structures combined with crosslinking steps supports the current controversial discussion if RNA-G4 exist or not. If this studies attempts to show the in vivo G4 landscape on the RNA level and wants to proof G4 existence, data has to be presented showing that crosslinking and treatment with two ligand is not affecting global processes in the cell.
2. It is easy to understand the next experiments, which are possible using this technique, however for publication at least one experimental strategy has to be presented.
3. The biological question is very interesting, and the authors have performed the first experiments to shed light on the topic. However, if RNA-G4 folded by ligands are no longer target of a given protein (normally unfolding proteins) the questions remain, which proteins (so far not known) proteins do the job? We know that this question maybe is beyond the topic of this research paper, but it is essential to state so.
4. Why is a two-step IP method needed? Meaning, how much G4s are induced by using two different G4 ligands in one assay? Can the method not be done using only the biotin-coupled ligand as a stabilization and a tag?
5. The TASQ ligands sounds very interesting. They have designed these ligand, but detailed analysis of this ligand is essential. Does this assay reveal similar data with ligands used extensively by others?
6. No raw data are listed or available (GEO fastQ files), this is essential to evaluate the data and sequencing analysis. Why was the sequencing coverage so poor? And why was the sequencing of the library not simply repeated on a HiSeq, which is widely available and much better suited for transcriptome analyses.
7. A detailed statistical analysis of RNAseq data is essential. How many mRNAs were enriched plus ligand? How many of those harbored a conventional G4 motif? How many targets lack a G4 motif (background)? Are all targets G-rich? Where are the G4s located? Where all targets previously identified by others?
8. On p6 line 103 – 106 the authors describe experiments to reveal how TASQ enters the cell. They revealed, using microscopic staining's that Nucleolin is the shuttling protein. The Figure, presented in the Supplement, is rather weak and not convincing. Co-IP data are required to proof this assumption. Furthermore, it is not clear from the current manuscript why this information is needed to understand the ligand function

Minor points:

The colors in the figures are not easy to understand quickly, because red, orange and yellow can look pretty similar on a bad printer. Also the use of the colors (and patterns) in 1b,c is not helping an easy understanding.

The model presented in Figure 3 is in its current layout not contributing to any novel aspect. The fact that trapped G4s (by crosslinking and ligand-binding (2 ligands) can no longer act like wild-type RNA molecules (and lead to more signals) is not only expected, but, in my opinion, already widely accepted.

Reviewer #3:

Remarks to the Author:

This manuscript presents experimental evidence supporting the existence of RG4s in living cells. Previous findings from Guo et Bartel (Nature, 2016) showed that RNA G-rich sequences are mostly unfolded in mammalian cells. However, they provided the indication that some RG4s are folded in vivo and a subset of them can be stabilized upon treatment with PDS. Using an in vivo cross-linking step to "freeze" RNA interactions, the authors here suggest that RG4s are transiently folded in vivo.

My major criticism is that they do not provide strong direct evidence of the "transient" nature of RG4 folding in cellulo. The G4RP-seq protocol should be also performed in the absence of FA. This is often used as a control but for this manuscript it could be important to investigate the notion of "transient" folding. Moreover, the authors excluded the possibility that BioTASQ modifies the RG4 landscape since FA cross-linking is performed before BioTASQ pull-down. However, this could occur in vitro during the G4RP protocol on noncrosslinked RNAs that are captured by the BioTASQ beads. Ectopically expressed reporters containing RG4s with different stability could also help in strengthen the notion of capturing transient RG4s. Fig. 2A and E: the capture of RNAs that do not contain RG4s should be included. As above, it would be important to show the enrichment without FA.

One important limitation of the protocol is that FA cross-linking captures direct/ indirect RNA-RNA as well as RNA-RBP interactions. The possibility that the RG4 pull-down could be mediated by protein factors (NCL?) should be controlled. Which RG4s are pulled-down? Inter or intra-molecular? Does the increase in G4 folding upon ligand addition occur on RG4s transiently folded or stable/formed RG4s? The release of DNAG4s that could also be pulled down should be controlled. Although the manuscript provides evidence that G4RP-seq captures G4-containing RNAs, the limitations of the G4RP protocol do not allow providing a complete assessment of the in vivo RG4 formation.

It would be important to validate I) RG4 formation on new RG4-containing RNA candidates II) the impact of the ligands on protein expression from RG4 transcripts.

Minor points:

The number of experimental repeats should be indicated.

The term "RIP" is misleading

NCOMMS-18-13424-A: Transcriptome-wide identification of transient RNA G-quadruplexes in human cells by *Yang et al.*

Detailed Response to Reviewers' Comments:

-----Reviewer #1 (Remarks to the Author):

This manuscript from Yang et al describes the detection of RNA G Quadruplexes in fixed cells. The authors employ G4RP-Seq - pulldown of G Quadruplexes followed by deep sequencing - to do so. This work is a timely contribution to a unsolved problem of key importance to RNA biology - do G Quadruplexes form in live eukaryotic cells?

This work fits nicely in a niche between live-cell G4-Seq (recent work from the Bartel lab and others), in which they observed these structures were unfolded in eukaryotes but folded in prokaryotes, and fixed-cell work with antibody staining from the Balasubramanian lab and others in which they observe G Quadruplexes in eukaryotic cells. Here, the Wong and Monchaud labs propose G4RP-Seq can be used as a snapshot-type method for capturing transient G Quadruplexes in eukaryotes. The field is currently in flux right now and this certainly won't be the final word on G4 RNA folding in eukaryotes, but that's precisely why this is an important contribution that will influence thinking in the field. It is suitable for the broad audience of Nature Communications with only minor revisions and one or two additional in vitro experiments.

Answer: We thank Reviewer #1 for the kind comments and confidence in our work.

My suggestions are below:

Major:

It is claimed that the pulldown complex is ternary (avidin/BioTASQ/G4 RNA). However, it's not clear to me that this is the case. Given the authors' previous observation in their report of N-TASQ that the 1:1 N-TASQ/TERRA complex can interact further with another equivalent of N-TASQ (e.g., Figure 2B of their 2015 JACS article) one would expect that at least a quaternary complex (1 avidin:2 BioTASQ:1 G4 RNA) is possible. Furthermore, this likely would only be possible in some cases, providing another source of bias in the experiments. Given the concentrations employed, it appears likely that the avidin molecules on the beads are loaded with multiple equivalents of BioTASQ, and quadruplexes that can interact with more than one BioTASQ at once will have longer bound lifetimes. I'm not insistent this is fully sorted out for a first report, but the authors seem to be most of the way there on an experiment that would be very helpful to readers. Doing some Job plots with soluble avidin-BioTASQ (or at least BioTASQ on its own) and a few G Quadruplexes of differing pulldown efficiency, and then correlating this to bound lifetime by binding fluorescently labeled quadruplexes to avidin beads with bound BioTASQ, washing, and monitoring the fluorescence of the supernatant over 2-3x the experimental timescale could provide some key information of value to readers in interpreting the data from this new technique.

Answer: We claimed that the complex is ternary because this is the minimal requirement for isolating quadruplex-forming sequence (QFS) *via* pull-down experiments. We agree that we do not have at present definitive structural demonstration of this contention. To gain insights, we performed stoichiometry investigations as requested. Classical techniques, such as job plot

analyses, are precluded here, given that the only chromophores in BioTASQ are guanine units, which are masked upon addition of increasing amounts of DNA/RNA (and *vice versa*). Instead, we performed ESI-MS investigations based on previous reports by V. Gabelica *et al.* (*Nucleic Acids Res.* **2002**, *30*, e82). Our results (now included as Figure S3A in the Supporting Information – see below) confirmed the low binding capability of BioTASQ (already discussed in the first version of our manuscript and in Figure S2) more illustratively since the 1:1 BioTASQ:G4 assembly was barely detectable. These results thus confirmed that the introduction of the biotin tag on the TASQ scaffold decreased G4-affinity of BioTASQ, thus making a comparison with previously reported TASQs with high G4-affinity (including N-TASQ) irrelevant. To further demonstrate this, we decided to directly compare G4-affinity between ^{PNA}DOTASQ (the parent TASQ without biotin) and BioTASQ *via* FRET-melting investigations. Our data (newly added as Figure S3B) confirmed that the introduction of biotin negatively impacted the ligand's G4-interacting properties, as shown by the introduction of a large excess of BioTASQ (5 μ M, 25-fold excess) imparted a low G4 stabilization (7°C), which was roughly half the measured G4 stabilization imparted by 1 μ M ^{PNA}DOTASQ under similar conditions. While BioTASQ exhibited a weaker G4 affinity compared to DOTASQ, its G4-selectivity remained high (Figure S2), making BioTASQ well-suited for *in vitro* pull-down experiments. BioTASQ's lower G4 affinity as compared to the parent compound is presumably originated from the steric contribution of the biotin tag that may negatively affect proper TASQ/quadruplex interaction, and this also explained why we decided to use a large excess of this probe for both the *in vitro* and *in vivo* experiments. In addition, we also performed (as requested) dose-response and time-dependent experiments; our data (summarized and added as the new Figure S3C) showed that TASQ concentration and the incubation time positively correlated with G4 sequence pulldown efficiency, thus justifying why we performed both the *in vitro* and *in vivo* pull-down experiments overnight. Finally, to gain further insights into the identity of the possible higher-order pulldown complexes, we performed *in vitro* pulldown in the presence of increasing amounts of BRACO-19 (newly added as Figure S3D), with both DNA and RNA quadruplexes, and showed that higher-order quaternary complexes (*i.e.*, avidin/BioTASQ/G4 RNA/BRACO-19) are possible. We hope that this series of additional control experiments satisfies Reviewer #1's expectations. Of note, Dr. Pauline Lejault, who performed this set of additional *in vitro* experiments, has been added to the author list.

Figure S3. A) Electrospray ionization mass spectra (ESI-MS) analysis that showed the low affinity and stoichiometry (1:1) of association between Myc quadruplex and BioTASQ. B) Confirmation of the lower quadruplex-affinity of BioTASQ as compared to the parent molecule ^{PNA}DOTASQ. FRET-melting experiments were performed with F-Myc-T (0.2 μM) with increasing amounts of ligands (1 and 5 μM, *i.e.*, 5 and 25 mol. equiv.). C) Pull-down experiments were performed with F-Myc (1 μM) and increasing BioTASQ concentrations (10 and 20 μM) and incubation times (4 and 16h). D) Competition BioTASQ (20 μM) pull-down experiments were performed with F-Myc and F-NRas (1 μM), and increasing amounts of BRACO-19 (0.5 and 1 μM).

Given the bias of the ligand employed for parallel G quadruplexes, some discussion (at least a more explicit acknowledgement to the reader) that this method may be missing some of the (rarer, but now known) antiparallel RNA G Quadruplexes would be helpful to those unfamiliar with the field. Similarly, a discussion of their experience with TASQ type ligands exhibiting tighter binding to G quadruplexes with exposed quartets that lack occluding loops would be helpful in allowing the reader to manage expectations regarding the limitations of this technique.

Answer: We agree; the manuscript has now been modified accordingly (page 5):

“The preference for BioTASQ for type I G4, which displays an accessible external G-quartet, is expected, given that TASQs are sterically demanding ligands that require accessible, loopless G-quartets for binding G4 targets efficiently. This represents a limitation to the use of BioTASQ for G4 detection, especially for G4-DNAs, which have higher conformational diversity than G4-RNAs. We have not yet tested the affinity of BioTASQ on the newly reported antiparallel G4-RNA. (29) While the cellular prevalence of G4-RNA with antiparallel topology remains to be established, they may provide key insights into the topological preference of TASQ ligands.”

Minor:

Concentrations are not clear. Oligonucleotides are reported as being present at 1e-6 M, but ctDNA is reported as being present at 100e-6M. Is this phosphate? Base pair?

Answer: We apologize for the omissions, the unit (“in base pairs”) has now been added twice in the manuscript (pages 6 and 17).

Page 2, line 23: this seems to conflate two phenomena. While many RNA G Quadruplexes are more stable than their DNA equivalents, this is a separate phenomenon from the fact that, in a cell, DNA is generally present paired to its complement, while RNA is single-stranded (or folded). Language should be adjusted.

Answer: We agree completely, and the manuscript has now been modified accordingly (page 2):

“While both single-stranded DNA and RNA can form G4s, the latter is less studied, (5,6) even though G4 formation in RNA molecules is generally more stable, and RNA molecules can fold more readily due to their predominant single-stranded nature in vivo.”

-----Reviewer #2 (Remarks to the Author):

In the manuscript “Transcriptome-wide identification of transient RNA G-quadruplex in human cell” by Yang et al. the authors claim the discovery of a novel technique to identify RNA-G-quadruplexes in vivo. RNA G-quadruplexes have attracted a lot of attention in the last years. Recently, a publication of the Bartel group in Science revealed that under their experimental conditions RNA-G4s are mainly unfolded. To address the controversial discussion, if G4 structures do not exist at all or if they are just rapidly unfolded by protein in the cell the authors attempted to identify RNA-G4s in vivo. Overall, the manuscript is well written, but the outlined experiments and results are not well presented and lack biological or medical relevance.

Answer: We thank Reviewer 2 for her/his comments. In our revised manuscript, we have divided the G4RP-seq results into two sections: baseline G4 transcriptome and ligand-induced G4 transcriptomic changes to reduce confusion. To streamline our presentation and focus the discussion on the transcriptome-wide profiling of G4 entities, data from our cellular imaging experiments were consequently removed. In the revised manuscript, BioTASQ was used exclusively as a probe rather than as a G4 ligand (further discussed below).

Major points

1. The experimental approach is well designed and interesting. However, it is not clear how an artificial ligand that is stabilizing RNA-G4 structures combined with crosslinking steps supports the current controversial discussion if RNA-G4 exist or not. If this study attempts to show the in vivo G4 landscape on the RNA level and wants to proof G4 existence, data has to be presented showing that crosslinking and treatment with two ligands is not affecting global processes in the cell.

Answer: While BioTASQ is a ligand itself, in this study, we used BioTASQ exclusively as a probe to capture G4-RNA targets much like a G4-specific antibody. The chemical crosslinking step was

performed on live cells to “freeze” or immobilize molecular structures and interactions before cell lysis and the *in vitro* incubation step with BioTASQ. We postulated that the chemical crosslinking step would keep the samples closer to an *in vivo* state and also minimize the stabilization effects of BioTASQ during pull-down protocol *in vitro*. To demonstrate this, we performed BioTASQ control experiments comparing RNA pull-down efficiency using crosslinked and non-crosslinked MCF-7 extracts (Figure S7, see below). In agreement with our predictions, overall higher levels of RNA were captured with BioTASQ using non-crosslinked samples, suggesting that *in vitro* incubation of naked (non-crosslinked) RNA with BioTASQ induced G4 formation, which was then captured by the streptavidin purification steps. We also observed a loss of signal differentials between BioTASQ-enriched (ie., HDGF, CCNI, KIAA0100) and BioTASQ-depleted genes (ie., MALAT, XIST, RPPH1) in non-crosslinked samples, suggesting that the BioTASQ incubation could induce *in vitro* G4 formation that could, in turn, masked the detection of *in vivo* G4 formation. Our data thus unambiguously confirmed the importance of chemical crosslink in the G4RP protocol to detect *in vivo* G4s. These additional data were incorporated and discussed in the manuscript (page 7):

“G4RP was performed in samples that were crosslinked to preserve the transiently formed cellular G4s while preventing the induction of G4 formation in vitro, as crosslinked nucleic acids would be in an immobilized state. As controls and to illustrate the importance of this crosslinking step, G4RP was performed in non-crosslinked samples. We selected three targets from the top- and bottom-ranked transcripts obtained with our G4RP-seq results (see below). BioTASQ enrichment of these targets was quantified using RT-qPCR and compared between the non-crosslinked and crosslinked samples (Figure S7). We observed a loss of difference in BioTASQ enrichment in the non-crosslinked samples, despite the overall higher signals. These increased signals in non-crosslinked samples were likely G4s that were formed in vitro, due to the high concentration of BioTASQ, arguing that the crosslinking step is necessary to immobilize cellular RNA structures and to minimize the effects of in vitro G4 formation and destabilization through the biochemical evaluation steps. As naked, non-crosslinked RNA targets are susceptible to the in vitro G4 stabilization effects of high concentration of BioTASQ during the incubation steps, the true differences between the in vivo transient levels of the top and bottom ranking targets would be masked.”

Proliferating MCF-7 cells were incubated with G4 ligands (BRACO-19 and RHPS4 at IC15 and IC25, respectively) for 72hr before formaldehyde crosslink and lysate extraction. Even with these sub-lethal treatment doses of G4 ligands, we fully expected ligand treatments to affect global cellular processes including the modulation of gene expression at the transcriptional level. To normalize for these ligand-induced transcript level changes, we included an input control for every condition such that each sample in our BioTASQ capture experiment was normalized internally (see Figure S6). We performed chemical crosslink with a standard concentration of formaldehyde (1%) and a 5-minute incubation step such that global biological processes should be fixed at the time of cell harvest. We contend that this normalization step is essential and sufficient to allow for an unbiased evaluation of G4-formation in the presence and absence of G4 ligand stabilization.

Figure S7. BioTASQ enrichment of G4-RNA using crosslinked versus non-crosslinked samples. RT-qPCR measurements of top ranked genes (HDGF, CCNI, KIAA0100) and lowest-ranked gene (MALAT1, XIST, RPPH1) from the G4RP-seq dataset at baseline (with no G4 ligand treatments) condition were tested. Error bars represent SEM.

2. It is easy to understand the next experiments, which are possible using this technique, however for publication at least one experimental strategy has to be presented.

Answer: We agree that it would be invaluable to combine G4RP-seq with another transcriptome-wide technique, and outlined this discussion in our manuscript (page 15):

“While using G4RP-seq alone cannot distinguish which competing factors (i.e. RNA-binding proteins or competing secondary structures) played more important roles at specific sites, data from G4RP-seq could be studied in combination with other functional genomic strategies (i.e. G4-ChIP-seq (60), RIP-seq (30), rG4-seq (9), DMS-seq (10), LIGR-seq (61)) to better characterize the interactions between G4 genome and transcriptome (collectively the G4ome).”

However, we believe that these additional investigations would be outside the scope of our current study. Such combinations of technique warrant extensive experimental optimization for both the laboratory techniques and the development of informatics analysis pipeline to answer a more complex and specific biological question. In this article, we seek to answer the most basic but pressing questions: whether transient G4-RNAs exist in the human transcriptome and how treatments with G4 ligands affect global G4-RNA landscape. We are currently under discussion to enter into an international partnership to probe the possibility of integrating different G4 RNA study platforms, and future studies will be aimed at combining G4RPA-seq with other techniques, [REDACTED].

3. The biological question is very interesting, and the authors have performed the first experiments to shed light on the topic. However, if RNA-G4 folded by ligands are no longer target of a given protein (normally unfolding proteins) the questions remain, which proteins (so far not known)

proteins do the job? We know that this question maybe is beyond the topic of this research paper, but it is essential to state so.

Answer: Due to the text limit in our previous submission, the discussion on helicases involvement in the resolution of G4-RNA was limited in scope. We contended that while helicases activity is essential to unfold G4-RNAs enzymatically, subsequent (or parallel) association with RNA-binding proteins could restrict G4 refolding, thereby keeping the G4-RNA unfolded after the initial unfolding step. We have now recapitulated this very idea in our revised discussion (page 13):

“Further, we anticipate that the equilibrium of an RNA’s transient folded and unfolded states can be influenced by competing factors such as transcription rate of the RNA, the availability of helicases (50), the chaperone activities of RNA-binding proteins (51) and the structural stabilization by G4 ligands (52)”

However, as the reviewer correctly pointed out, the identity of many of these helicases and RNA-binding proteins are not currently known (see also Ref 10: Guo and Bartel, Science 2016). We hope that our technique could be optimized in the future to address these relevant and important biological questions.

4. Why is a two-step IP method needed? Meaning, how much G4s are induced by using two different G4 ligands in one assay? Can the method not be done using only the biotin-coupled ligand as a stabilization and a tag?

Answer: We sincerely apologize for any misunderstandings. Our protocol is in fact based on a single IP step. To clarify our approach, we have now consequently removed all experiments involving the use of BioTASQ as a G4 ligand from the manuscript (mostly optical imaging studies, which were part of the former Figures 1 and S3). BioTASQ was used here exclusively as a probe in the G4RP-seq protocol, similarly to an antibody, in the single IP step of the protocol. Two G4 ligands (BRACO-19 and RHPS4) were used to treat live MCF-7 cells before the cells were chemically crosslinked and extracted cell lysate subjected to the G4RP protocol. G4RP-seq analyses of these treated samples then revealed critical information on how G4 ligand treatments changed the G4 transcriptome landscape. We selected these two ligands since they are well established and extensively studied. Both ligands readily crossed cellular membrane with known affinities against DNA and RNA-G4s. BioTASQ was used as an *in vitro* probe to capture transcriptome-wide RNA targets of these G4 ligands.

The G4RP protocol is based on a chemical crosslinking step to lock down the conformational changes induced by G4 ligand treatments (during treatments in live cells) and the use of BioTASQ to capture these immobilized G4s. As it stands, G4RP-seq could be used to study the treatment effects of any G4 ligand of interest in live cells, thus considerably expanding the scope of our approach. The discussion has now been completely rewritten (page 12) to provide a clearer message.

5. The TASQ ligands sounds very interesting. They have designed these ligand, but detailed analysis of this ligand is essential. Does this assay reveal similar data with ligands used extensively by others?

Answer: BioTASQ is the newest generation of TASQ ligand, after the development of prototypes including biomimetic (DOTASQ), smart (^{PNA}DOTASQ) and twice-as-smart G4 ligands (PyroTASQ and N-TASQ). The latter (N-TASQ) has been used in live cells to demonstrate the existence of both G4-DNA and G4-RNA in functional cellular context, based on a live-cell compatible optical imaging approach. The design of BioTASQ originates from the data collected over the years with TASQ probes (in terms of efficiency and selectivity), making it highly adaptable due to its biotin moiety that could be used in various imaging and pull-down protocols. As discussed in the text (page 4), BioTASQ interacts with G4 *in vitro* in a manner that is similar to the previously reported TASQ (notably regarding G4-specificity) albeit with a lower G4-affinity than that of the parent ^{PNA}DOTASQ (discussed in the Supporting Information, Figure S3B). However, comparing these properties between BioTASQ and other live-cell G4 trackers reported so far would be difficult given their intrinsically different cellular behaviours (including bioavailability, biostability, as well as the different modality of detection).

6. No raw data are listed or available (GEO fastQ files), this is essential to evaluate the data and sequencing analysis. Why was the sequencing coverage so poor? And why was the sequencing of the library not simply repeated on a HiSeq, which is widely available and much better suited for transcriptome analyses.

Answer: As required by Nature Publishing Group, we have deposited the raw sequence data onto GEO (accession number: GEO112898), which is included in the Data Availability section of our manuscript. Some of our samples displayed lower sequencing depth than the others. When comparing the high sequence depth data sets with that of the lower-depth samples, we observed high reproducibility of the expression patterns in biological replicates, even though the sequencing datasets of these replicates had different sequencing depths. As we are convinced that even our low-coverage datasets provided adequate depth for the purpose of our analyses, all samples were normalized by sequencing depth for subsequent comparison with the remaining replicates. While this sacrifices coverage, the hierarchy of the biological data would still be intact, and our observations on G4-RNA level changes/events remained valid. In our accompanied bioinformatics analyses, we instead focused on the subset of the most abundant gene transcripts. We have also separately validated these events in selected genes, including lncRNAs MALAT1, XIST, and RPPH1 using qPCR. We contend that the sequencing depth used in our analyses is completely appropriate for the goal of this study, which is to demonstrate the validity of the G4RP-seq protocol and to provide a proof-of-principle on the presence of global G4-RNAs in the transcriptome. However, we are aware that future study aiming at the high-resolution mapping of RNA-G4 sites will need to be performed at a much greater depth. We discussed this limitation repeatedly in the manuscript (pages 9 and 15):

“While the sequencing depth was not high enough to detect subtle changes in individual G4-forming sequences, we were able to confidently determine gene-level changes by focusing on a subset of highly abundant transcripts”

“Additionally, the sequencing depth in our work was insufficient to differentiate small changes in lowly-expressed transcripts and individual G4-forming sequences, and we anticipate that high-resolution sequencing in future studies will be necessary for a more complete mapping of potential transcriptomic G4 sites”

7. A detailed statistical analysis of RNAseq data is essential. How many mRNAs were enriched plus ligand? How many of those harbored a conventional G4 motif? How many targets lack a G4 motif (background)? Are all targets G-rich? Where are the G4s located? Where all targets previously identified by others?

Answer: We sincerely thank Reviewer 2 for suggesting these bioinformatics analyses, and agree that a thorough study of our data set, by comparing them with different G4 motif prediction algorithms, was required. To this end, we added a comprehensive pG4 motif analyses on the G4RP-seq data to evaluate the baseline G4 transcriptome, as well as the ligand-induced G4 transcriptomic changes. Figures 3 and 4 (see below) now includes pG4 distributions of BioTASQ-enrichment ranked gene transcripts. In Figure 3, we showed the baseline G4 transcriptome and demonstrated a correlation between BioTASQ enrichment and pG4 distribution. We added the corresponding text in the results section (page 9):

“ES, a gene-specific ratio of the BioTASQ signal normalized to the corresponding input signal, moderately positively correlated with their respective G/C content (Pearson correlation=0.43, $p<0.0001$) (Figure 3b). ES was uncorrelated to gene length (Pearson correlation=0.02, $p=0.89$) (Figure 3c). To evaluate whether the ES is related to the density of potential G4 sequences, Quadbase2 (41) was then used to assess the number of predicted G4 (pG4) motifs (i.e. sequences with the canonical G3L1-7 G4 sequence) in each gene transcript. The ratios of the number of pG4s to gene length were calculated and then plotted against the transcripts ranked by their ES (Figure 3d). Comparison of the ratios between the top and bottom 100 ES-ranked transcripts showed significantly higher values (2.2-fold difference, $p<0.001$) for top-ranked transcripts. Results from gene ontology analysis of the top and bottom 100 ES-ranked transcripts are summarized in Table S1. (42) Together, our bioinformatics analyses confirmed that transcripts with higher ES tend to have higher G/C content and higher pG4 density.”

Figure 3. Characterization of the baseline level G4-RNA landscape using G4RP-seq. **a**, Top 10 highly abundant transcripts (filtered by at least 500 base-read counts) with the lowest BiotASQ enrichment (blue bars, normalized to the input in the untreated sample) or with the highest BiotASQ enrichment (red bars) ranked by Enrichment Scores. **b**, Regression plot of BiotASQ Enrichment Score for each transcript versus its corresponding G/C content. ($R^2=0.187$, $P<0.001$, significant non-zero relationship) **c**, Regression plot of BiotASQ Enrichment Score for each transcript versus its corresponding gene length ($R^2=0.00005$, $p=0.89$, non-significant relationship). **d**, Number of pG4 motifs (calculated by Quadbase2 using mid stringency G3L1-7) to gene length ratio plotted against the subset of highly abundant transcripts ranked by their BiotASQ enrichment. The bar graph on the top right show the average pG4 motif/gene length ratio between the top 100 ranked transcripts versus the bottom 100 ranked transcripts ($p<0.001$, significant difference).

Bioinformatics analyses of the ligand-induced changes revealed further insights into the differential G4 induction profiles of the two G4 ligands. We have now modified Figure 4 (see below) to include the new data and added the following text under the results section (page 10):

“We found no correlation between the ligand-induced ΔES and the read counts of the transcript within that list (Pearson correlation = 0.02, $p=0.69$) (Figure S8). The ligand-induced ΔES of each transcript was compared to the corresponding G/C content of the transcripts, which interestingly showed a negative correlation (Figure 4b). When the transcripts were ordered by ΔES , the pG4 density appeared to be distributed toward lower scores (1.7-fold and 2.7-fold difference between average ratio of top and bottom 100 ranked transcripts for BRACO-19 and RHPS4 respectively) (Figure 4b, bottom panel). Overall, our observations suggest that transcripts with higher pG4 density were more likely to be captured in a folded state in the absence of ligands, resulting in ΔES being lower due to the higher baseline level of G4 formation. In contrast, transcripts with lower pG4 density were more likely to be unfolded in the absence of ligands and to have their G4 structures stabilized in the presence of ligands, leading to a higher ΔES .

When we compared the absolute number of pG4 motifs (i.e. without normalization to gene length) between the three treatment conditions ranked by their respective ES, we observed differential changes in pG4 profiles between the two G4 ligand treatments (Figure 4c). pG4 scores generated using different stringency of searches (G2L1-10, G3L1-5, and G3L1-7) showed similar trends (Figure S9). BioTASQ-captured targets generated from BRACO-19-treated samples exhibited higher levels of pG4-dependent enrichment regardless of the search stringency, conceivably due to the broader range of intramolecular G4s (longer loops, 2-quartet G4s, etc.) stabilized by this ligand. On the other hand, targets generated from RHPS4-treated samples showed lower levels of pG4-dependent enrichment and a pG4-dependency could only be observed when the plots were obtained using G4 motif searches with the lower stringency. While we reasoned that RHPS4 binding preference could be selective towards sequences with lower numbers of G4 motifs, we cannot rule out the possibility of intermolecular G4s, as computational algorithms are currently unable to predict these structures. Given this caveat, the lack of pG4-dependent enrichment in samples treated with RHPS4 could be partially explained by a preferential ligand-induced stabilization of intermolecular G4s. Overlap between the gene lists for the two G4 ligand treatments demonstrated that they have differential G4-induction profiles, in agreement with their differential in vitro G4-structure-specific binding profiles (Figure 4d) (43).”

The supplementary excel files with the list of highly abundant gene transcripts and corresponding BioTASQ enrichment scores has now been updated to include the parameters for pG4 calculations and analyses (using Quadbase2) (see Supplementary File S1D).

8. On p6 lane 103 – 106 the authors describe experiments to reveal how TASQ enters the cell. They revealed, using microscopic staining's that Nucleolin is the shuttling protein. The Figure, presented in the Supplement, is rather weak and not convincing. Co-IP data are required to proof this assumption. Furthermore, it is not clear from the current manuscript why this information is needed to understand the ligand function

Answer: We agree. This part of the text was confusing (see also our answer to point 4 above) and the cellular imaging experiments did not contribute directly to the development of the G4RP protocol. These experiments and the corresponding text were consequently removed from the revised manuscript. [REDACTED].

Minor points:

The colors in the figures are not easy to understand quickly, because red, orange and yellow can look pretty similar on a bad printer. Also the use of the colors (and patterns) in 1b,c is not helping an easy understanding.

Answer: We agree. We replaced these similar colour spectra with new color scheme for Figure 1 (see below). The new Figures 3 and 4 were created with the new color scheme as well (see above).

Figure 1. Characterization of G4-specific affinity of BioTASQ. **a**, Structure of BioTASQ displaying a biotin affinity tag (red circles), and schematic representation of its open (left) and closed, quadruplex-associated conformation (right), in which the intramolecular G-quartet is formed. Schematic representation of a guanine-rich RNA sequence (guanines as grey squares) in its unfolded, random-coil and folded G4 structure. **b**, Fluorescence analysis of pull-down experiments carried out with 1) FAM-labeled oligonucleotides (1 μM): either G4-DNA (F-MYC, F-SRC and F-22AG), duplex-DNA (F-DS) or G4-RNA (F-TERRA, F-TRF2 and F-RAS); 2) BioTASQ (20 μM); and 3) streptavidin-coated magnetic beads. **c**, Competitive pull-down experiments performed with F-SRC and F-NRAS (1 μM), BioTASQ (20 μM) in the absence or presence of duplex-DNA competitors (ds17 or ds26, 20 μM) or DNA extracts (calf thymus DNA, ctDNA, 100 μM), or of a molecular competitor (biotin, 80 μM or ^{PNA}DOTASQ, 10 μM). All experiments were done in triplicates.

The model presented in Figure 3 is in its current layout not contributing to any novel aspect. The fact that trapped G4s (by crosslinking and ligand-binding (2 ligands) can no longer act like wild-type RNA molecules (and lead to more signals) is not only expected, but, in my opinion, already widely accepted.

Answer: We understand the reviewer's reservation. Our model (now as the new Figure 5), while speculative, is an interpretation of our findings in context with previously published observations. We contend that G4RP-seq provided evidence that G4-RNAs are formed as intermediates, albeit transiently, in RNA/RNP biogenesis. Furthermore, there has not been, until now, evidence of G4-ligand-mediated stabilization of transcriptome-wide G4-RNAs, in which we also included in our model. Chemical crosslink was used to immobilize G4s and thereby allowing for the capture of these proposed transient structures in various biological processes, as well as quantifying their presence as endpoint measurements. Using this protocol, we also successfully demonstrated that ligand-induced stabilization of G4-RNA could also be quantified, and that individual G4-ligands exhibited G4-structure specific affinity to native G4-RNA targets in human transcriptome, in agreement with oligonucleotide-based *in vitro* screening studies. Finally, G4RP-seq provides a much-needed quantitative assay for building an empirical data-based algorithm to predict G4-ligand RNA-binding landscape, an essential tool to aid the rapid development of G4-ligand as therapeutics against human diseases. We thus believe that the schematic representation now seen in Figure 5 is an important visual support for the non-specialist readers of *Nature Comm.* to better understand the implications of our findings and the scope of the G4RP-Seq protocol. We hope that the Reviewer will agree with us.

-----Reviewer #3 (Remarks to the Author):

This manuscript presents experimental evidence supporting the existence of RG4s in living cells. Previous findings from Guo et Bartel (Nature, 2016) showed that RNA G-rich sequences are mostly unfolded in mammalian cells. However, they provided the indication that some RG4s are folded in vivo and a subset of them can be stabilized upon treatment with PDS. Using an in vivo cross-linking step to “freeze” RNA interactions, the authors here suggest that RG4s are transiently folded in vivo.

Answer: We thank reviewer #3 for her/his comments.

My major criticism is that they do not provide strong direct evidence of the “transient” nature of RG4 folding in cellulose. The G4RP-seq protocol should be also performed in the absence of FA. This is often used as a control but for this manuscript it could be important to investigate the notion of “transient” folding. Moreover, the authors excluded the possibility that BioTASQ modifies the RG4 landscape since FA cross-linking is performed before BioTASQ pull-down. However, this could occur *in vitro* during the G4RP protocol on noncrosslinked RNAs that are captured by the BioTASQ beads.

Answer: As demonstrated in our previous replies (notably to point#1, Reviewer 2), we are fully aware of the possibility that BioTASQ could affect the G4-RNA landscape. We postulated that the chemical crosslink step before BioTASQ introduction should minimize the *in vitro* effects of BioTASQ as an inducer of G4-RNA formation, mentioned here. To directly answer Reviewer 3’s queries, we now included in our supplementary section, comparison data between non-crosslinked vs. crosslinked G4RP experiment (Figure S7 and below). Please refer to Q1 of Reviewer#2 for more details.

Ectopically expressed reporters containing RG4s with different stability could also help in strengthen the notion of capturing transient RG4s. Fig. 2A and E: the capture of RNAs that do not contain RG4s should be included. As above, it would be important to show the enrichment without FA.

Answer: While ectopically expressed RG4 reporter could be used as controls, we elected to remain focus on biological relevant targets, with their native/endogenous copy numbers (as opposed to over-expression and abnormally high copy number in the case of artificial reporter expression). We hope that the Reviewer will agree with our strategy.

As discussed in the previous answer, our data showed a clear differential between BioTASQ-captured G4-RNA signals from transcripts with high RG4 potential when compared with signals from transcripts with low RG4 potential, but only when the samples were chemically crosslinked to immobilize the endogenous structures. This differential was lost in non-crosslinked samples, suggesting that the high BioTASQ concentration used in the *in vitro* IP step may induce G4 formation in naked RNA post-extraction.

One important limitation of the protocol is that FA cross-linking captures direct/ indirect RNA-RNA as well as RNA-RBP interactions. The possibility that the RG4 pull-down could be mediated by protein factors (NCL?) should be controlled. Which RG4s are pulled-down? Inter or intra-molecular? Does the increase in G4 folding upon ligand addition occur on RG4s transiently folded or stable/formed RG4s? The release of DNAG4s that could also be pulled down should be controlled. Although the manuscript provides evidence that G4RP-seq captures G4-containing RNAs, the limitations of the G4RP protocol do not allow providing a complete assessment of the *in vivo* RG4 formation.

Answer: We agree with the Reviewer that these are all true limitations of the G4RP-seq protocol, and have now expanded our discussions on these limitations in our revised manuscript (page 15):

“The G4RP-seq protocol has certain limitations and, we anticipate, can be further improved. One concern is that BioTASQ is itself a G4 ligand and thus may alter the G4 landscape; however, because the chemical crosslink step before the BioTASQ binding step minimizes the effects of BioTASQ-induced stabilization, the protocol should provide a relatively unbiased readout. BioTASQ preference toward parallel G4s could limited its capability in capturing rarer forms of G4-RNAs (ie. antiparallel G4s). Also, the protocol uses formaldehyde as cross-linking agent, and this is known to capture both direct and indirect RNA-RNA interactions; (59) it will be informative in future work to also include the use of other cross-linking agents to better characterize the G4-RNA interactome. Additionally, the sequencing depth in our work was insufficient to differentiate small changes in BioTASQ enrichment in lowly-expressed transcripts and individual G4-forming sequences, and we anticipate that high-resolution sequencing in future studies will be necessary for a more complete mapping of potential transcriptomic G4 sites. Despite these limitations, we have shown here that G4RP-seq is useful for the identification of transient G4 structures and offers a novel snapshot view of the G4 landscape in live human cells.”

At this stage of our method development, we cannot determine the precise nature and the extent of G4 interactions (RNA-RNA, protein-RNA, protein-protein). It is our intent with the publication of this protocol, that we and other researchers in the field, will further optimize the G4RP protocol with the use of different cross-linking agents, alternate capture probe/antibodies, and using different experimental conditions or systems. Datasets from these parallel investigations could be integrated then for more comprehensive analyses of the RNA-G4 landscape and the complexities of their molecular nature.

Chemical crosslink with formaldehyde was chosen over other agents as its crosslink efficiency is based on the proximity of targets. Arguably, this close-range crosslink property means that indirect interactions should be minimal. Our crosslink step is also short (5 minutes), so we do not expect over-crosslinking leading to aggregations and/or stochastic non-biologically relevant interactions.

Ligand-mediated G4-stabilization effects are mostly likely directed at nascent RNA since the report by Guo and Bartel (Science 2016, Ref. 10) showed evidence that are generally unfolded at steady-state, presumably when they were at their functional-state. We showed in our analyses that the distribution of pG4s is correlated to the baseline G4 levels. However, transcripts with the lower distribution of pG4 motifs were found to harbour higher Δ ES, suggesting that they were better induced by G4 ligands. We provided our reasoning to explain this apparently paradoxical observation in our text, as well as in our reply to Reviewer 2’s query above (please refer to Reviewer 2 point #7 for more detail). We have also included a section in the discussion section (page 13) regarding the ligand effects:

“Following ligand treatments, we found, paradoxically, that transcripts with lower pG4 density were relatively more enriched than those with a higher number of G4 motifs. There are two possible explanations for this. First, the relative increase in BioTASQ

enrichment is less significant in high-pG4 transcripts than that found in low-pG4 transcripts, which have, by definition, a lower probability of being crosslinked in a folded state in the absence of ligands. Upon ligand treatment, stabilization of these previously unfolded pG4s within the low-pG4 transcripts significantly increased their BioTASQ enrichment. A second possibility is that rather than the density of G4 motifs, ligand-induced G4 stabilization may be more dependent on the ligand-binding affinity of individual G4 sequences, taking into account other confounding factors such as topology, flanking or loop sequences, and the presence of modified nucleosides. This is supported by our observation that the absolute number of pG4s within a transcript seemed to be more predictive of its ligand-induced G4-structure formation for BRACO-19, a broad spectrum, pan-G4-specific ligand, but not for RHPS4, a ligand with higher structural and sequence specificity. Given that the G4-binding modes of BRACO-19 and RHPS4 are different, their pG4 profiles are expected to be different.”

It is not currently feasible to fully determine whether the targets are intramolecular and/or intermolecular G4s. We tried not to over-extrapolate from the pG4 profiles of the dataset, stating only that “(...) *the absolute number of pG4s within a transcript seemed to be more predictive of its ligand-induced G4-structure formation for BRACO-19, a broad spectrum, pan-G4-specific ligand, but not RHPS4, a ligand with higher structural and sequence specificity*” (see above).

Finally, the BioTASQ-capture samples were DNase-treated before sequencing and qPCR were performed with exon-spanning primers, so G4-DNAs should not confound the results. While our article does not provide a complete assessment of G4 transcriptome, our goal is to report on the methodological advances and proof-of-principle to achieve this ultimate goal.

It would be important to validate I) RG4 formation on new RG4-containing RNA candidates II) the impact of the ligands on protein expression from RG4 transcripts.

Answer: Interesting point. To further validate G4 targets reported by our G4RP-seq protocol, we selected Quadruplex-forming sequences (QFS) from the top 3 genes (MALAT1, XIST and RPPH1) through the analysis of the transcripts sequences using QGRS Mapper. RG4 formation of these new candidates was confirmed using established *in vitro* techniques including circular dichroism (CD) and thermal differential spectra (TDS). These new data were added in the revised manuscript (page 11) and in the Supporting Information (see below):

“Circular dichroism (CD) and thermal differential spectra (TDS) analyses (47) of the three selected pG4 regions of MALAT1, XIST and RPPH1 further confirmed the formation of parallel-type G4 structure in vitro in the selected pG4 motif sequences extracted from these genes (Figure S11).”

Various published works have shown that G4 ligand treatments can induce changes in the translation of specific RNA-G4-containing transcripts (e.g., Cammas et al., *RNA Biology* 2015 and *Nucleic Acid Res.* 2017 (Ref. 7 here), 2015; Katsuda et al., *J. Am. Chem. Soc.* 2016), and

demonstrated translational regulation through RNA-G4 level modulations. To demonstrate the global effects of G4-RNA formation, western blots analyses on the changes at the protein level of individual targets would not provide substantive data that can be generalizable transcriptome-wide. To properly investigate the effects of transcriptome-wide RNA-G4 formation on translation control, parallel large-scale proteomics studies will need to be performed. While it is interesting to evaluate the effects of G4-RNA formation on protein expression, it is perhaps outside the scope of our study.

Figure S11. Confirmation of quadruplex topology by circular dichroism (CD) and thermal differential spectra (TDS) investigations of selected quadruplex-forming sequences (QFS) of top ranked gene (MALAT1: r[G₃ATG₃AG₂AG₅TG₃] (G-score: 40); XIST: r[G₂AAG₂AAG₂TTG₂] (G-score: 21); and RPPH1: r[G₂AG₄CCCG₂CG₂] (G-score: 21)). Typical G4-RNA parallel quadruplex signatures were obtained by both CD (positive (260-265nm) and negative peaks (240nm)) and TDS (positive (250-270nm) and negative peaks (295-300nm), dashed arrows).

Minor points:

The number of experimental repeats should be indicated. The term “RIP” is misleading

Answer: We have added these omissions where they are appropriate (in the figure legends and methods section). While we agree that technically, our probe-based purification scheme did not qualify as an immuno-precipitation experiment, RIP is a commonly accepted term for RNA or RNP purification. We have taken care to ensure that the readership will understand that we did not use any antibodies in our purification scheme.

Reviewers' Comments:

Reviewer #1:

Remarks to the Author:

In this revision from Yang et al, the authors have addressed my concerns with new in vitro data better characterizing the nature of the interaction of their BioTASQ ligand with G-Quadruplexes. Suggested changes to the text have been made as well. Experiments and changes to the text responsive to the requests of the other reviewers have also been made.

I recommend publication with one minor change: The CD and TDS results given in response to reviewer 3 are presented with a little too much certitude. These are fingerprint-type spectra that follow rules of thumb, but the collective experience of the RNA community with antiparallel RNA G Quadruplexes is minimal, and there's reason to think the rules might be different. For example, Spinach, which contains an antiparallel quadruplex, has a CD spectrum more like a parallel one, in Chem Commun (Camb). 2015 May 28; 51(43): 9034–9037. This admittedly is convolved with the dsRNA content of the aptamer's contribution to the spectrum, but I think it gives reason for caution. I support including this data with the change of the phrase in the added p11 text from "further confirmed" to "were consistent with."

Reviewer #2:

Remarks to the Author:

Dear Authors, I am very pleased with the changes in the manuscript and I have no further concerns or questions.

Reviewer #3:

Remarks to the Author:

Despite a thorough response from the authors to the referees' remarks, I still believe that the G4RP-seq method allows the identification of G4 structures in cellulo (with the limitations discussed below) but does not provide conclusive evidence of the "transient" nature of these structures. The crosslinking step indeed allows freezing the interactions in cellulo and the comparison with a sample not treated with FA might have shown that G4s are detectable only when stabilized by FA, thus demonstrating that RNA G4s are formed in cellulo only in a transient manner. The new figure S7 shows that the G4RP-seq method does not allow this analysis since the ligand coupled to biotin induces post lysis G4 folding of the naked RNA in the untreated sample. The hypothesis of a transient formation in cells is very relevant and could change our understanding of the role and regulation of these structures in living cells but in my opinion this is not clearly demonstrated in this manuscript. Moreover, Fig S7 (which lacks statistics) definitely shows that the cross-link step is important to capture G4 formed in cellulo but also indicates a possible bias in data interpretation: if the efficiency of the crosslink is not 100%, the naked RNA fraction present in the sample treated with FA will structure in vitro and the biotin coupled to BiotTASQ will capture RNA G4s formed in cellulo but also those structured in vitro after lysis. I wonder if doing an RT-qPCR on the sample treated with FA before cross-link reversion could control the presence of naked non-crosslinked RNA. It is important to test this possibility to be able to claim that this method allows studying the landscape of G4-RNA in cellulo. In addition, since FA cross-links also proteins to RNA and, given the results on nucleolin, I still wonder whether the pull-down is mediated by RBPs binding G4 RNA (i.e. indirect capture of RNAs). Proteinase K treatment might show that recruitment of G4 RNA is not dependent on protein factors.

Our revised manuscript has been re-examined by 3 referees and we are delighted that all of them agreed on the timely relevance and the significance of our work. We are happy to see that both Reviewers #1 and #2 were satisfied by the revisions that were made in response to their comments. While Reviewer #3 agreed that our results are of utmost importance for the field (« *the hypothesis of a transient formation in cells is very relevant and could change our understanding of the role and regulation of these structures in living cells* »), the Reviewer expressed reservations on our conclusion on the transient nature of G4-RNA formation, and would not recommend publication in *Nature Communication* at this stage. Specifically, the Reviewer asked for two additional control experiments: first, a series of RT-qPCR validations without crosslink reversal step (that « *could control the presence of naked non-crosslinked RNA* »), and second, a series of RT-qPCR validations after proteinase K (PK) treatment (that « *might show that recruitment of G4 RNA is not dependent on protein factors* »). We respectfully disagree with this Reviewer (with the support of the positive reviews from Reviewers #1 and 2) and strongly believe that the burden of proof for the transient existence of G4-RNA in human cells has been met. Our responses are listed below in line with the Reviewer's comments (**in bold**):

The new figure S7 shows that the G4RP-seq method does not allow this analysis since the ligand coupled to biotin induces post lysis G4 folding of the naked RNA in the untreated sample. Moreover, Fig S7 (which lacks statistics) definitely shows that the cross-link step is important to capture G4 formed in cellulo but also indicates a possible bias in data interpretation: if the efficiency of the crosslink is not 100%, the naked RNA fraction present in the sample treated with FA will structure in vitro and the biotin coupled to BiotTASQ will capture RNA G4s formed in cellulo but also those structured in vitro after lysis.

G4RP includes a crosslinking step to mitigate an expected *in vitro* G4-induction bias. As shown in Figure S7, this crosslink is crucial to discriminate the differential between transcripts of different pG4 density *in vivo*. This is also reflected by our thorough bioinformatics analyses of G4RP-seq data supporting that the extent of G4 formation of an RNA, in the absence of G4-ligand, is positively correlated to the pG4 density of the RNA. Additionally, while we agree that the crosslink efficiency would not be 100%, our FA concentration and incubation time are similar to the standard protocol in ChIP-seq experiments. Under these experimental conditions, the current estimate for DNA crosslink efficiency is 90% (*J. Biol. Chem.* **2015**, *290*, 26404). While we did not find similar data for cellular RNA, we expected the FA-mediated RNA crosslink efficiency should even be higher, considering that the kinetics should be more efficient in the cytoplasmic compartment. Most importantly, if non-crosslinked, naked RNAs contribute substantially to the G4RP-signals (*i.e.*, through BioTASQ-induced G4-formation during the *in vitro* incubation step), then it would be impossible to detect significant ligand-induced effects as *in vitro*, BioTASQ-induced G4 formation of naked RNA would dominate the G4RP signals. This is clearly not the case.

I wonder if doing an RT-qPCR on the sample treated with FA before cross-link reversion could control the presence of naked non-crosslinked RNA. It is important to test this possibility to be able to claim that this method allows studying the landscape of G4-RNA in cellulose.

We did attempt the G4RP protocol without crosslink reversion during our protocol optimization phase, without much success. The omission of crosslink reversal disrupts RNA extraction, leading to major biases. Conceivably, as most cellular RNAs were crosslinked under our experimental conditions, they would be co-purified with the organic/macromolecule fractions and discarded in the RNA extraction step. The poor quality of RNA extraction affects subsequent qPCR quantification. In addition, performing RT-qPCR directly on crosslinked lysate is not recommended due to many uncontrollable factors (including the crosslinked RNA itself) that can affect reverse transcription and PCR.

In addition, since FA cross-links also proteins to RNA and, given the results on nucleolin, I still wonder whether the pull-down is mediated by RBPs binding G4 RNA (i.e. indirect capture of RNAs). Proteinase K treatment might show that recruitment of G4 RNA is not dependent on protein factors.

Bioinformatics analyses of G4RP-captured RNA targets showed that the enriched transcripts contain high potential G4-forming motif (pG4) density. If the major driver of the G4RP protocol is indirectly through protein binding, then we should detect a background level of cellular RNPs that should not show any detectable dependency on pG4s.

As well, the suggested use of proteinase K is not compatible with the G4RP protocol. Proteinase K digestion requires high incubation temperature (usually at 55°C or higher) to degrade cellular polypeptides. At this temperature, we also expected a partial reversal of crosslink, and the end effects of reduced G4RP-signal could be misinterpreted. Moreover, the crosslinking step itself affects proteinase K digestion efficiency. If a high concentration of enzyme were used to overcome this limitation, the digested sample then would need to be additionally purified before RT-qPCR, and this uncontrolled additional purification step would defeat the entire purpose of the proposed experiment. We are cognizant of the possibility that a low percentage of G4RP capture could be indirect and have already addressed this in our discussion as a limitation of the protocol.

Accordingly, in this revised manuscript, we added Reviewer 1's editorial suggestion, included the statistic test in Supplementary Figure 7, as suggested by Reviewer 3; and added/reiterated our discussions on the limitations of our approach, to address Reviewer #'s concerns. For a detailed list of all the changes we made, please refer to the track-change version of our revised text.